# An Empirical Study on the Application of TDA to Deep Neural Networks

## Abstract

This study aims to analyze the global structure of the functional subgraph of DNNs using tools from topological data analysis (TDA), namely persistent homology (PH) and the Betti curve similarity. Using these methods we present an empirical study on the application of TDA to DNNs in order to gain a better understanding of their architecture and to provide a framework for a similarity measure between DNNs. The study is conducted by training several convolutional neural networks (CNNs) on disjoint subsets of the ImageNet dataset and then by analyzing the structure of their functional graphs across datasets using the Betti curve similarity. Results show that the Betti curve similarity is able to distinguish between different DNN models across datasets and can be a tool for detecting a departure from previous internal representations of those datasets, providing a new method for the analysis of DNNs and a potential path forward for their theoretical development.

## 1 Introduction

With the seemingly ubiquitous implementation of deep neural network (DNN) algorithms in modern applications, it has become increasingly important for scientists and practitioners of deep learning, to develop methods for the analysis and scrutability of these algorithms. There have already been various attempts, and small triumphs, with tools such as SHAP values, LIME and XNN (Agarwal and Das, 2020), to name a few, but a complete framework for the scrutability of DNNs has yet to emerge. The sheer size of these DNNs is one of the major reasons why they remain inscrutable, and recent trends seem to indicate that DNNs will only become larger, thus exacerbating this problem.

For these particular reasons we show that a candidate tool for analyzing the global structure of DNNs is persistent homology (PH) and its corresponding summary statistic, the Betti curve. Both of these originate from topological data analysis (TDA), a branch of abstract topology composed of tools for computing the global structure of data. To demonstrate their uses in deep learning, we modify and add upon work by Corneanu et al. (2019) by analyzing the functional graphs of convolutional neural networks (CNNs) and comparing those graphs across time, i.e. epochs, and datasets. This is done by first training several distinct CNNs on disjoint datasets, extracting their activations on the respective testing data, reducing the activation data via a $k$-means++ algorithm, processing this reduced data using persistent homology, and finally comparing our results using their respective Betti curves, as seen in Figure 1.

## 2 Methods

The proposed analysis begins with training a series of CNNs on disjoint subsets of the ILSVRC2017 dataset (Russakovsky et al., 2015), commonly known as ImageNet. The global structure of the CNNs' functional graphs across datasets and epochs are then analyzed using PH and the Betti curve similarity. For reproducibility, the details of the data, the CNN models, and the TDA tools that are used in the study along with the random seed are provided. The code used for the study is largely a modification of the previous work by Corneanu et al. (2019), which provides the scaffolding for the models, data loaders, training, and activation extraction. Our modifications are available here at GitHub and the data is available for download at ImageNet. All of the packages used in the study are listed in the REQUIREMENTS.TXT file in the GitHub repository.

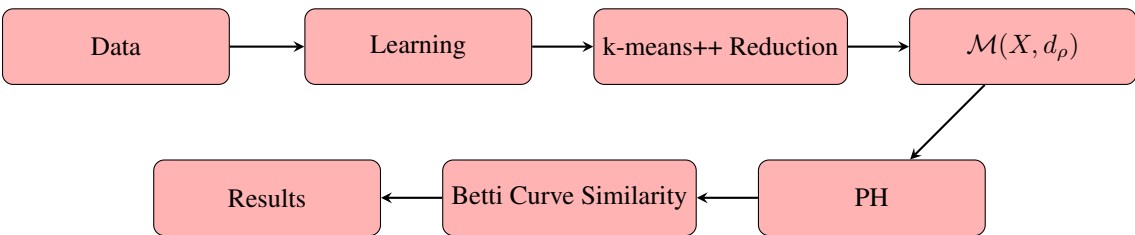

Figure 1: Flowchart of the study where we (1) train several distinct CNNs on disjoint datasets, (2) extract their activations on the respective testing data, (3) reduce the activation data via a $k$-means++ algorithm, (4) process this reduced data using persistent homology, and (5) comparing our results using their respective Betti curves.

Throughout the study the PyTorch library is used for the implementation of models, datasets, data transformation, and data loaders. The study is conducted on a supercomputing cluster utilizing a single node with two 64-core AMD EPYC 7763 (2.45 GHz) processors with 512 GB of RAM each, and two NVIDIA A100 GPUs with 80 GB of memory each. A full experiment on a given dataset, excluding training, utilizing seven epochs for computation, averages 66 minutes across models.

## 2.1 DATA

The machine learning task studied here is image classification over ten categories. In order to compare CNN models across subsets of data and epochs, a dataset is required that is large enough to be divided into several subsets (in our case 30 in order to provide a statistically significant sample size without incurring excessive computational overhead) of training and test instances. For its size, availability and ease of use, ImageNet is a natural choice.

The original ImageNet training dataset consists of 1.2 million images, each of which is labeled with one of 1000 categories. The original test dataset consists of $50,000$ images, each of which is also labeled with one of its respective categories and where each category contains 50 images. The training dataset is balanced so that each category is represented by the same number of instances, in this case 732 (this being the number of samples in the category with the fewest number of images).

Thirty disjoint subsets of ten categories each are randomly selected using seed 1234; these subsets are then held constant throughout the entire experiment in order to compare the CNN models across the different subsets. Because the data set contains images of varying heights and widths, every image is resized to be $64 \times 64$ pixels. All images are then standardized by subtracting the mean and dividing by the standard deviation of the respective training subset. This ensures that the results are consistent with standard practice and that no data leakage occurs between the training and test sets. At training time, the subsets are augmented by randomly flipping the images horizontally and adding random color jitter in order to help prevent overfitting and improve the generalization of the models.

## 2.2 TRAINING

Four different CNN models are trained across all of the subsets: an extended LeNet model, an AlexNet model, a VGG-16 model, and a ResNet-18 model. This allows us to compare the global structure of the CNNs' functional graphs across the different models, subsets, and epochs. Each of the model's architectures are essentially the same as their original counterparts with the exception of the extended LeNet model, which has an additional two linear layers. This is done to increase the accuracy of the extended LeNet model and enable better comparability between it and the other models. As expected, the extended LeNet model performs the worst out of the four models, with the ResNet-18 model performing the best, in terms of accuracy. Figure 2 shows the average accuracy of each of the models across the different subsets. Note that the models are trained using the same hyperparameters and optimizer settings, which are detailed below.

During training subsets are randomly sampled using a batch size of 100. Each of the models is trained using the Adam optimizer with a learning rate of 0.001 and a weight decay of 0.0005. Cross-entropy

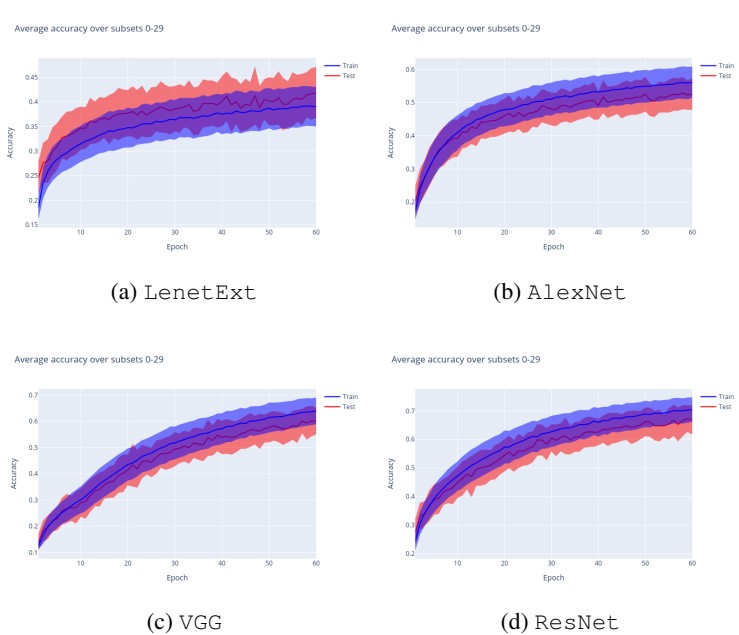

Figure 2: Average accuracies of our models over all subsets.

loss is used to calculate the loss between the predicted and actual labels, and training lasts for 60 epochs, with model weights saved for epochs $0, 10, 20, 30, 40, 50$, and $60$.

## 2.3 Non-linear Dimensionality Reduction

In order to make the analysis more computationally feasible, the number of neuron activations from each of the layers of the CNNs is reduced using a PyTorch GPU accelerated $k$-Means++ algorithm (Omer, 2020). This reduction allows the construction and PH computation of the functional graphs in a reasonable amount of time. $k$-Means++ was chosen as the reduction technique due to its non-linear nature and its previous success in reducing the dimensionality of point clouds for PH analysis (Malott and Wilsey, 2019).

In order to construct the functional graphs of the CNNs for a given subset, neuron activations are extracted from each of the layers of the network for each of the images in the test set by passing the images (transformed by the corresponding training set transform) through the CNNs and then extracting the neuron activations from each of the layers. The activations are then stored in an array of size $M \times N$, where $M$ is the number of images in the test set and $N$ is the aggregated number of neuron activations from each layer. Any neuron activations whose variance is zero are discarded, since these activations do not contribute to the global structure of the functional graph due to the fact that the correlation between them and other neuron activations is always zero. The neuron activations are then prepared for the $k$-Means++ algorithm through standardization.

To reduce the number of the neuron activations for each model, the $k$-Means++ algorithm was used to cluster the neuron activations into $1000$ clusters[1]. Dimensionality reduction is effected by replacing each neuron activation in a cluster with the neuron activation which is closest to the cluster's centroid. These neuron activations are then used as the reduced set for the PH analysis, which allows us to construct the functional graphs of the CNNs. Analysis of the silhouette scores for the clusters for each of the models show that the clusters were poorly separated, which in turn shows that the means of the neuron activations are not well-separated.

---

[1]Given the limitations on our computational resources and the complexity of the PH analysis, $1000^2$ activations is the largest number of points that we currently can feasibly analyze using PH.

This reduction in the number of the neuron activations introduces some approximation error into the analysis. However, we argue that the $k$-Means++ algorithm is able to capture the global structure of the neuron activations in a non-linear way. It has also been argued that the local structure of the neuron activations is not as important as the global structure, since the local structure is more representative of overfit in the model (Corneanu et al., 2019). Therefore, we believe that the $k$-Means++ reduction is a suitable method for the analysis of the global structure of the functional graphs of the CNNs, which is the main focus of our study.

## 2.4 FINITE METRIC SPACES AND FUNCTIONAL GRAPHS

A **finite metric space** is a finite set of points $X$ equipped with a function $d_X : X \times X \to \mathcal{R}^+$ that satisfies the properties of a metric, i.e., non-negativity, symmetry, and the triangle inequality. Since the set of points is finite, we can also completely describe the metric by a distance matrix $D_X$ where $(D_X)_{i,j} = d_X(x_i, x_j)$ for all $x_i, x_j \in X$. In this way, we can represent the finite metric space as a weighted graph, where the vertices are the points of the metric space and the edge weights are the distances between the points.

We formalize the functional graph of a DNN as a finite metric space, where the points are given by the neuron activations of the DNN and the distance between two activations is given by

$$d_\rho(\mathbf{a}_i, \mathbf{a}_j) = \sqrt{1 - |\rho(\mathbf{a}_i, \mathbf{a}_j)|} \tag{1}$$

where $\rho(\mathbf{a}_i, \mathbf{a}_j)$ is the correlation between the neuron activations $\mathbf{a}_i$ and $\mathbf{a}_j$. We note that the distance function $d_\rho$ satisfies all properties of a metric except for positivity, since $d_\rho$ equaling 0 does not imply that the inputs are the same (López De Prado, 2016). Further, $d_\rho$ is satisfied by several different correlation functions, such as the Pearson and the Spearman correlation. For our study, we use the Spearman correlation as our correlation function $\rho$, since it is able to capture both linear and non-linear relationships and does not require that the neuron activations be normally distributed (Kutner, 2005).

In order to construct the weighted graph representing the functional graph of a given network `net`, we first took its reduced set of neuron activations from 2.3 and constructed its distance matrix $D_{\texttt{net}}$ using equation 1. With the distance matrix $D_{\texttt{net}}$ we then calculated the persistent homology of the functional graph of `net`.

## 2.5 TOPOLOGICAL DATA ANALYSIS

Topological data analysis is a framework for analyzing the underlying topological space of a given dataset. It comprises a suite of tools from abstract topology used to construct and count the combinatorial objects which model the structure of topological spaces. In our case, we use the Giotto-tda library (Tauzin et al., 2020) to calculate the PH of the functional graphs of the CNNs and extract the Betti curves from their persistence diagrams. The Betti curves are then used to calculate the Betti curve similarity between the CNN models across the different subsets and epochs. We provide a quick overview of terminology and concepts from TDA from which we derive the necessary tools (Edelsbrunner and Harer, 2010).

Let $d$ be a positive integer and let $\{x_0, x_1, \ldots, x_n\} \subset \mathbb{R}^d$ be a finite set of points. An $n$-**simplex** is the convex hull of $n+1$ affinely independent points, often denoted by $\sigma = [x_0, x_1, \ldots, x_n]$ where the dimension of $\sigma$ is $n$. A **face** of $\sigma$ is any of the simplices of equal or lesser dimension that are contained in $\sigma$ and is often denoted by $\tau \leq \sigma$. The **boundary** of $\sigma$ is the union of all proper faces of $\sigma$ where a **proper face** is simply a face of strictly lesser dimension, denoted $\tau < \sigma$. A **simplicial complex** $K$ then, is a finite collection of simplices such that if $\sigma \in K$ and $\tau \leq \sigma$, then $\tau \in K$, and if $\sigma_1, \sigma_2 \in K$, then $\sigma_1 \cap \sigma_2$ is a face of both or is empty.

Let $K$ be a simplicial complex and let $C_n(K)$ be the free abelian group generated by the $n$-simplices of $K$. The objects of $C_n(K)$ are called $n$-**chains** and are formal sums $c = \sum_{i=1}^n a_i \sigma_i$ where $a_i \in \mathbb{Z}_2$ and $\sigma_i$ is an $n$-simplex of $K$. For any two elements $c_1, c_2 \in C_n(K)$, addition is defined similarly to that of polynomials, i.e., $c_1 + c_2 = \sum_{i=1}^n (a_i + b_i)\sigma_i$. Thus, the $n$-chains of $C_n(K)$ form a vector space over $\mathbb{Z}_2$ and are known as **chain groups**.

The **boundary operator** $\partial_n : C_n(K) \to C_{n-1}(K)$ is a linear map between chain groups. It takes as input an $n$-chain $c = \sum_{i=1}^{n} a_i \sigma_i$ and maps it to the $(n-1)$-chain $\partial_n c = \sum_{i=1}^{n} a_i \partial_n \sigma_i$. The boundary operator operates on the $n$-simplex $\sigma = [x_0, x_1, \ldots, x_n]$ by

$$\partial_n \sigma = \sum_{i=0}^{n} (-1)^i [x_0, \ldots, \hat{x}_i, \ldots, x_n] \tag{2}$$

where $\hat{x}_i$ denotes the removal of the $i$-th vertex of the simplex; essentially taking an $n$-chain and sending it to its boundary.

Given a simplicial complex $K$ and a dimension $p$, the $p$-th boundary operator $\partial_p$ is used to define what are known as **cycles** and **boundaries** of the chain group $C_p(K)$, written $Z_p(K)$ and $B_p(K)$, respectively.

A $p$-chain $c \in C_p(K)$ is a cycle if $\partial_p c = 0$ and is a boundary if there exists a $(p+1)$-chain $b \in C_{p+1}(K)$ such that $\partial_{p+1} b = c$. This means then that $Z_p(K) = \ker \partial_p$ and $B_p(K) = \operatorname{im} \partial_{p+1}$, making them both subspaces of $C_p(K)$. Further, due to properties of the boundary operator, it turns out that $B_p(K) \subseteq Z_p(K)$; therefore, the quotient space

$$H_p(K) = Z_p(K)/B_p(K) \tag{3}$$

is defined, and is known as the $p$-**th homology group** of $K$. It is essentially the span of the $p$-cycles which are also not boundaries, and it is used to describe the topological structure of the complex.

In order to construct the PH of a given dataset, we construct its corresponding complex iteratively by adding in simplices a few at a time. This is known as a **filtration** of the complex and must satisfy

$$\emptyset = K_0 \subseteq K_1 \subseteq \cdots \subseteq K_n = K, \tag{4}$$

where the indices are dependent on a filtration parameter which is often treated as a time scale.

Given a simplicial complex $K$ and a filtration $\emptyset = K_0 \subseteq K_1 \subseteq \cdots \subseteq K_n = K$, the **persistent homology** of $K$ is a measure of the scale of the topological features throughout the filtration, and homology groups of the complex are tracked as the filtration progresses. For a given dimension $p$ and indices $i \leq j$, the $p$-th persistent homology group of $K$ is defined as

$$H_p^{i,j}(K) = Z_p(K_i)/(B_p(K_j) \cap Z_p(K_i)), \tag{5}$$

with the $p$-**th persistent Betti number** of a simplicial complex $K$ defined as $\beta_p^{i,j} = \operatorname{rank} H_p^{i,j}(K)$. The Betti numbers of a simplicial complex are used to count the number of $p$-dimensional generators of space and therefore give a unique summary (up to isomorphism) of its topological structure. It is with these that we construct our Betti curves which allow us to compare networks.

The **Vietoris-Rips complex** $V_\epsilon$, is a simplicial complex that is used to approximate the topology of a finite metric space by constructing simplices from its points. Given a finite metric space $(X, d)$, the simplices of $V_\epsilon(X)$ are the subsets of $X$ whose diameter is less than or equal to the filtration parameter $\epsilon$, and where the diameter is defined to be the maximum distance between any two points in the subset. The complex is then constructed from these simplices.

This is done by starting with the points of $X$ as our $0$-simplices and connecting them with edges if the pairwise distance between them is less than or equal to $\epsilon/2$. We get higher and higher dimensional simplices by continuing to increase $\epsilon$ and adding more pairwise intersections between the points. Creating new simplices however comes at a cost, as the youngest simplices are the first to be removed while the eldest live on. This is known as the **Elder Rule** and we say that a simplex is **born** at the filtration parameter $\epsilon_i$ and **dies** at $\epsilon_j$ if it is added to the complex at $\epsilon_i$ and removed at $\epsilon_j$.

For our study we use the multi-threaded Vietoris-Rips complex implementation from Giotto-tda to compute the PH of the finite metric space of each of our functional graphs $D_{\texttt{net}}$. This particular implementation has been shown to be efficient and scalable for large datasets (Tauzin et al., 2020), even outperforming certain `C++` and `GPU` accelerated implementations.

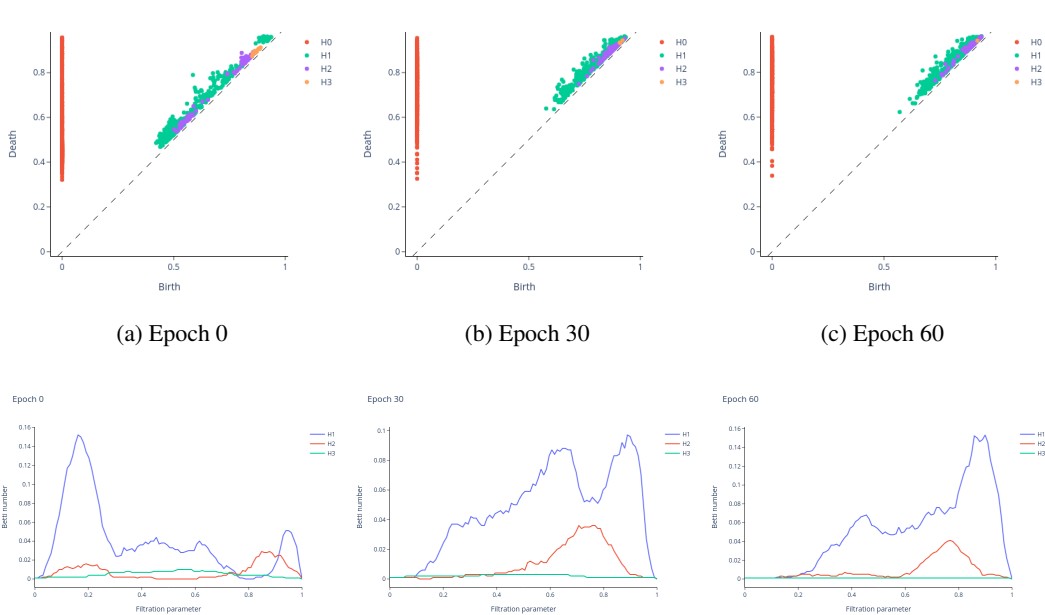

Figure 3: Persistence diagram of the reduced functional graph of ResNet-18 at epochs 0, 30, and 60 for homology dimensions 0–3 and their corresponding Betti curves.

The **persistence diagram** $\mathcal{P}_{\text{net}}$ of our Vietoris-Rips complex $V_\epsilon(D_{\text{net}})$ is a visual representation of the Betti numbers of the complex as a function of the filtration parameter, and fully encodes the information from the PH of the complex. As seen in Figure 3, the persistence diagram is a plot of the birth and death times of the topological features of the complex. A feature is considered to be persistent if its death time is reasonably larger than its birth time, and is considered to be noise otherwise, i.e., if it is close to the diagonal. The persistence diagram is then used to calculate the corresponding **Betti curves** according to

$$\beta_{\text{net}}^p(\epsilon) = \left| \{ \mathbf{x} \in \mathcal{P}_{\text{net}}^p \,|\, x_1 < \epsilon \le x_2 \} \right| \tag{6}$$

where $\mathcal{P}_{\text{net}}^p$ is the subset of the persistence diagram for the $p$-th persistent homology group of the complex (Edelsbrunner and Harer, 2010), and where $\epsilon \in [0, 1]$. An example can be seen in Figure 3.

After having calculated the Betti curves $\beta_{\text{net}}^p$ for each of the CNN models across the different subsets and epochs, we calculate the Betti curve similarity between the models. As far as we are aware this is the first time that the Betti curve similarity has been used to compare the global structure of DNNs across datasets and epochs.

The **Betti curve similarity** in dimension $p$ is computed by simply taking the infinity norm of the difference between the Betti curves of two models, i.e.,

$$\text{BCS}_p(\texttt{net}_1, \texttt{net}_2) = \left\| \boldsymbol{\beta}_{\texttt{net}_1}^p - \boldsymbol{\beta}_{\texttt{net}_2}^p \right\|_\infty \tag{7}$$

## 3 RESULTS

Here the results of the study are presented for training four different CNN models across 30 disjoint subsets of the ImageNet dataset and analyzing the global structure of their functional graphs using PH and the Betti curve similarity. The Betti curve similarity is able to capture the differences in the global structure of the CNNs' functional graphs across the different models, subsets, and epochs. The

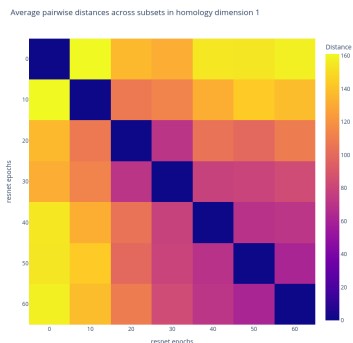

Figure 4: Average Betti curve similarity across all subsets of the ResNet-18 model with itself for homology dimension 1.

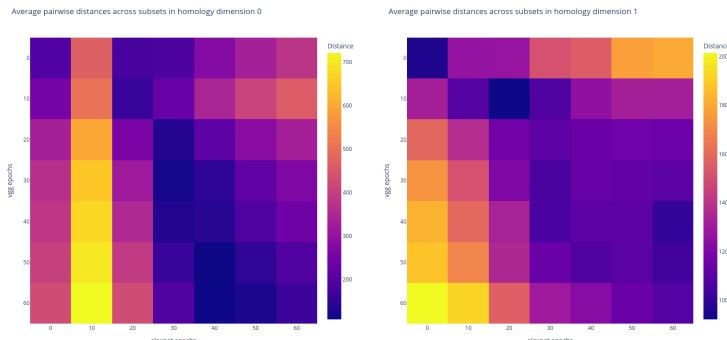

Figure 5: Average Betti curve similarity across all epochs of AlexNet compared to VGG-16 for homology dimensions 0 and 1.

most interesting results of the study are highlighted here along with a discussion of their implications. Additional results can be found in the supplemental material A.1.

### 3.1 FUNCTIONAL SIMILARITY ACROSS TIME

Comparing the average unnormalized similarity over time reveals that temporal similarity is quite low at the beginning of training and then typically increases as the models learn the features of the dataset. For example, in Figure 4 the similarity between the ResNet-18 model at epoch 0 and the same model at epoch 60 is quite low, indicating that the global structure of the functional graphs of the network changes over the course of training. A large shift in similarity from epoch 0 to epoch 10 is also evident, where the accuracy of the model is increasing most rapidly. Both of these are to be expected as the network learns the features of the dataset, as seen in Figure 2. Further, in Figure 4, the convergence of the network's functional graph towards some global structure can be observed, as the similarity between adjacent epochs is increasing. The same phenomenon appears over the other persistent homology dimensions as well (Figure 17). Also of interest is the fact that, on average, the similarity between the models seems to be increasing when compared at the same epoch. This is especially true in the zeroth and first persistent homology dimensions, and can be seen in Figure 5 in which AlexNet and VGG-16 are compared, hinting that the global structures of the functional graphs of the models are becoming more similar as the models are trained and that perhaps on average the models are converging towards the same global structure (Mao et al., 2024).

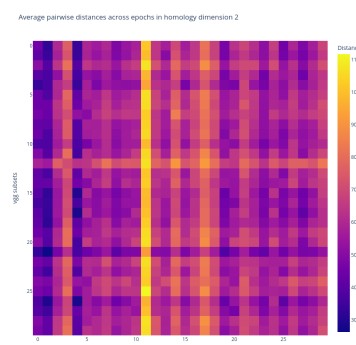

Figure 6: Average similarities over all epochs of ResNet compared to VGG-16 for persistent homology dimension 2.

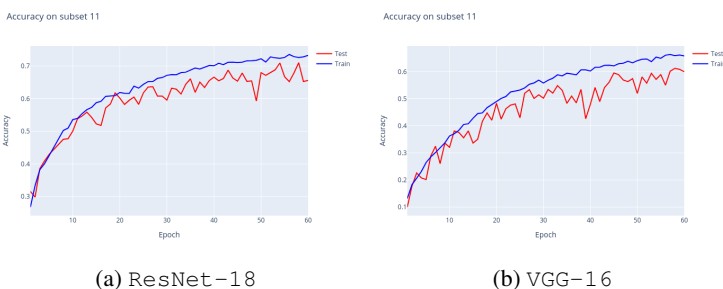

(a) `ResNet-18`         (b) `VGG-16`

Figure 7: Train and test set accuracies on subset 11 for ResNet-18 and VGG-16.

## 3.2 FUNCTIONAL SIMILARITY ACROSS DATA

Comparing the average unnormalized similarities across the different subsets of testing data, reveals that the models' functional graphs are quite dissimilar, i.e., the models' representation of the subsets are not the same. For certain models and subsets, the similarity was quite low, indicating that the representation for that particular subset was quite different from the others and that the models seem to be representing the features of the dataset in different ways. For example, in Figure 6 it can be seen that the similarity between the ResNet-18 model and the VGG-16 model for subset 11 is very low. Further inspection of the subset itself reveals that the classes in the subset are very distinct (see subsubsection A.1.5) as compared to others (e.g., subset 25 with three classes of dog), and the accuracy of the models on subset 11 shown in Figure 7 reveals that ResNet-18 outperforms VGG-16 by approximately 5% on the testing set . It can be further observed that for subset 27, a subset with similar classes in terms of morphism, the similarity between the models ResNet-18, VGG-16 and AlexNet was quite high, while they all differ considerably from the extended LeNet model as seen in Figure 8. Looking at the accuracy of the models on this subset however, would not readily reveal this difference, as the models' performance in terms of accuracy are all distinct, with the extended LeNet model performing most poorly, with AlexNet coming in second worst, as seen in Figure 9. Therefore, it can be concluded that statistically the models are creating distinct internal representations of the testing data across subsets. This is somewhat surprising since the models are not fundamentally different, being simply CNNs of varying sizes, but this also evidences that a simple change to the architecture topology, namely the residual connections in ResNet, can make a large difference for certain datasets.

## 4 CONCLUSION AND FUTURE WORK

We have introduced some theoretical tools from TDA for analyzing the global functional structure of deep neural networks and have shown that the Betti curve similarity can be a useful tool for the

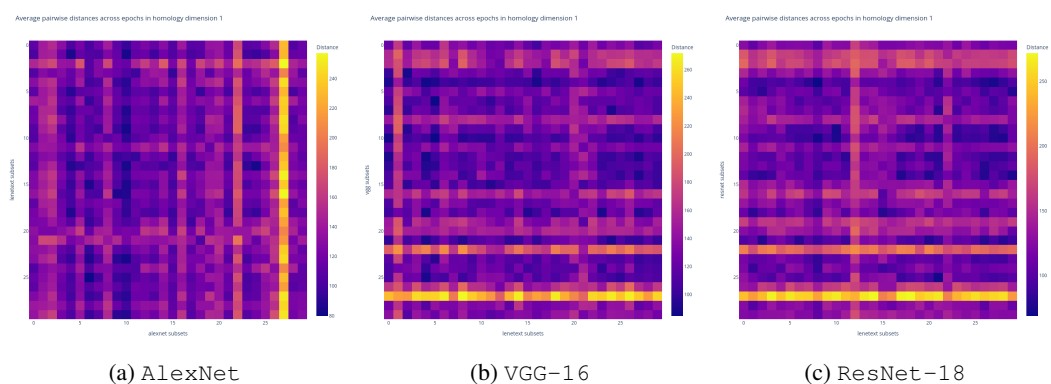

(a) `AlexNet`  (b) `VGG-16`  (c) `ResNet-18`

Figure 8: Average similarities over all epochs of LenetExt compared to all other models in homology dimension 1.

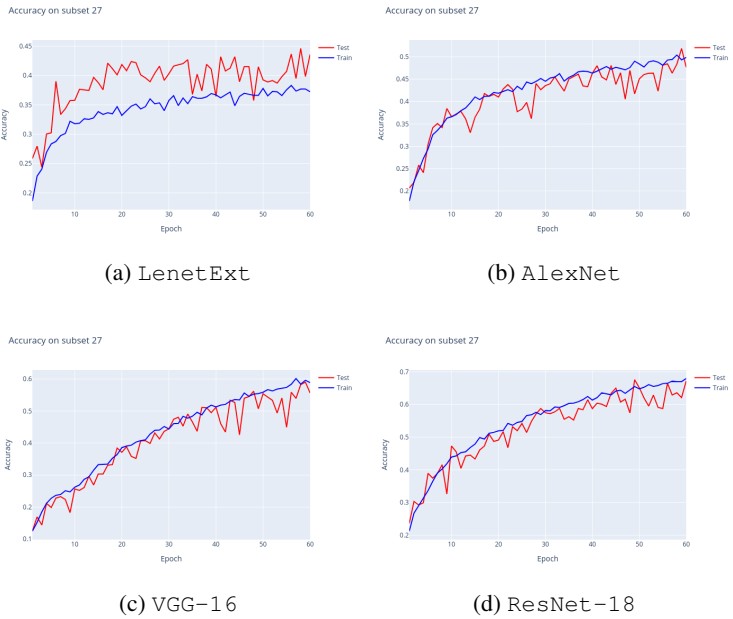

(a) `LenetExt`  (b) `AlexNet`

(c) `VGG-16`  (d) `ResNet-18`

Figure 9: Training and test accuracies for each model on subset 27.

comparison and analysis of DNNs. As a companion to accuracy and other metrics, the Betti curve similarity can provide a more nuanced understanding of the architecture and training dynamics of DNNs, and could be utilized in ablation studies and hyperparameter tuning. Thus, these tools may allow for more intentional creation of DNNs instead of the current ad hoc approach. We demonstrate some of the potential uses of the Betti curve similarity in our study by analyzing the functional graphs of CNNs. However, it is likely that this approach can be used in the analysis of other machine learning models and other types of data. Further, there are likely many more applications to which it may be applied, including model engineering, model compression, and transfer learning.

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

# A APPENDIX

## A.1 SUPPLEMENTAL MATERIAL

### A.1.1 SELF SIMILARITY ACROSS TIME

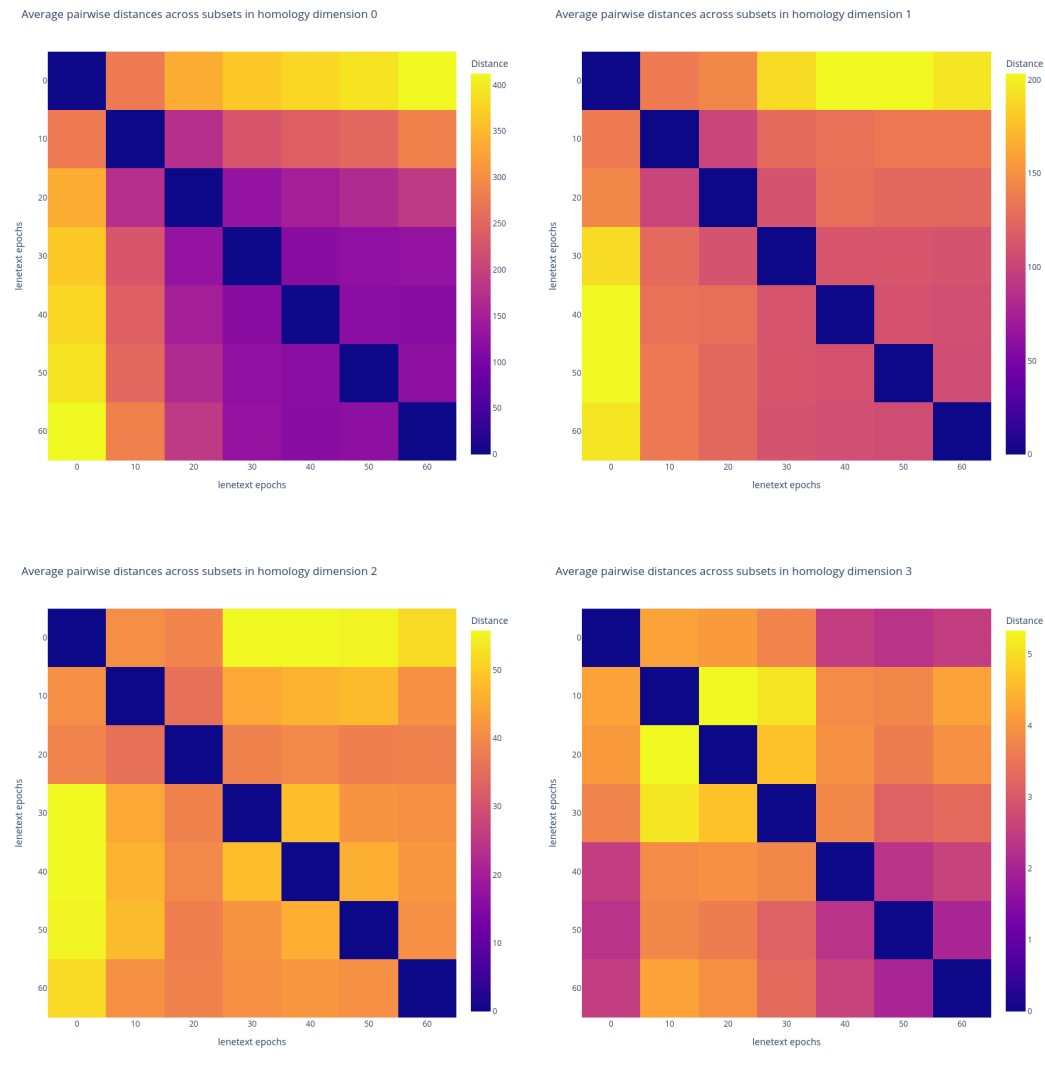

Figure 10: Average similarities over all subsets of LenetExt for each persistent homology dimension.

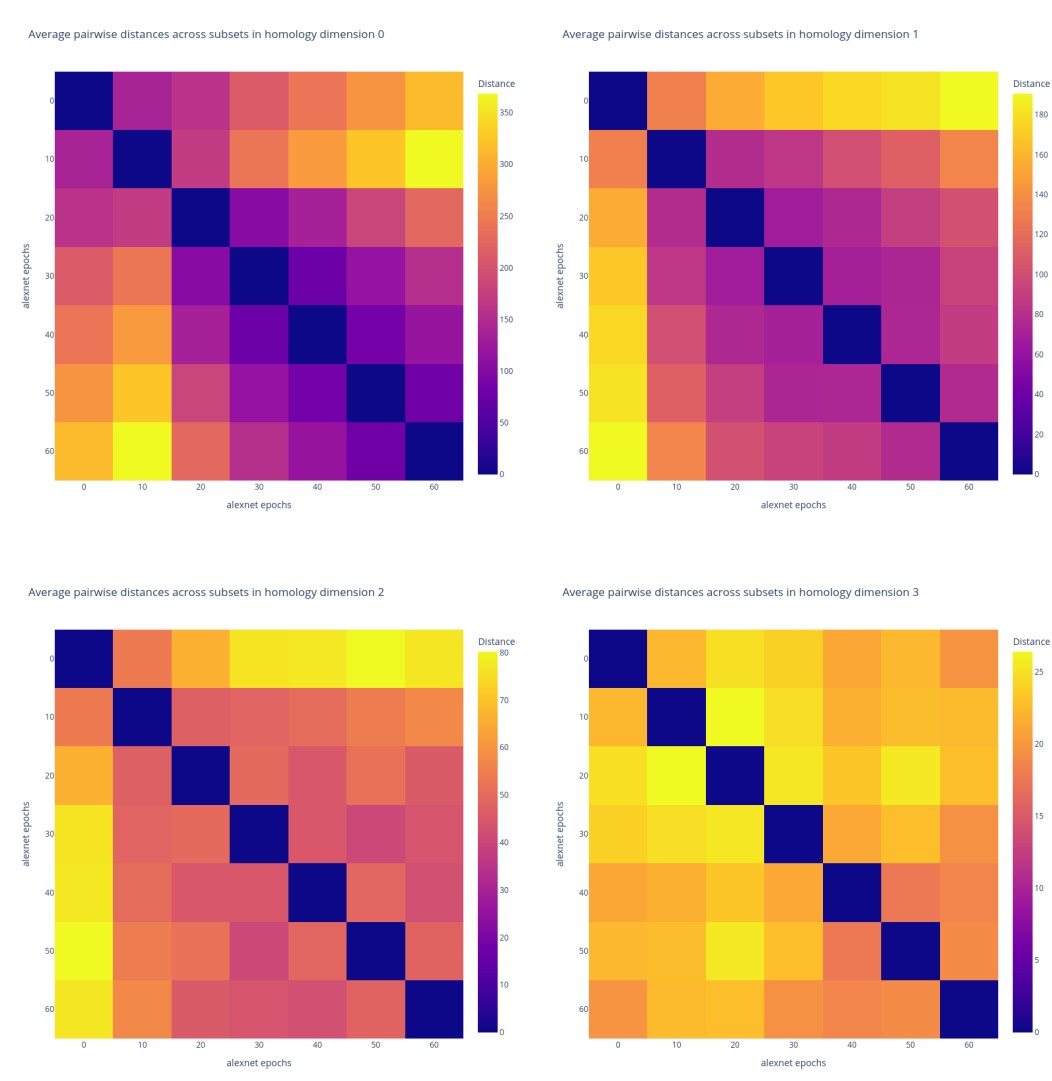

Figure 11: Average similarities over all subsets of AlexNet for each persistent homology dimension.

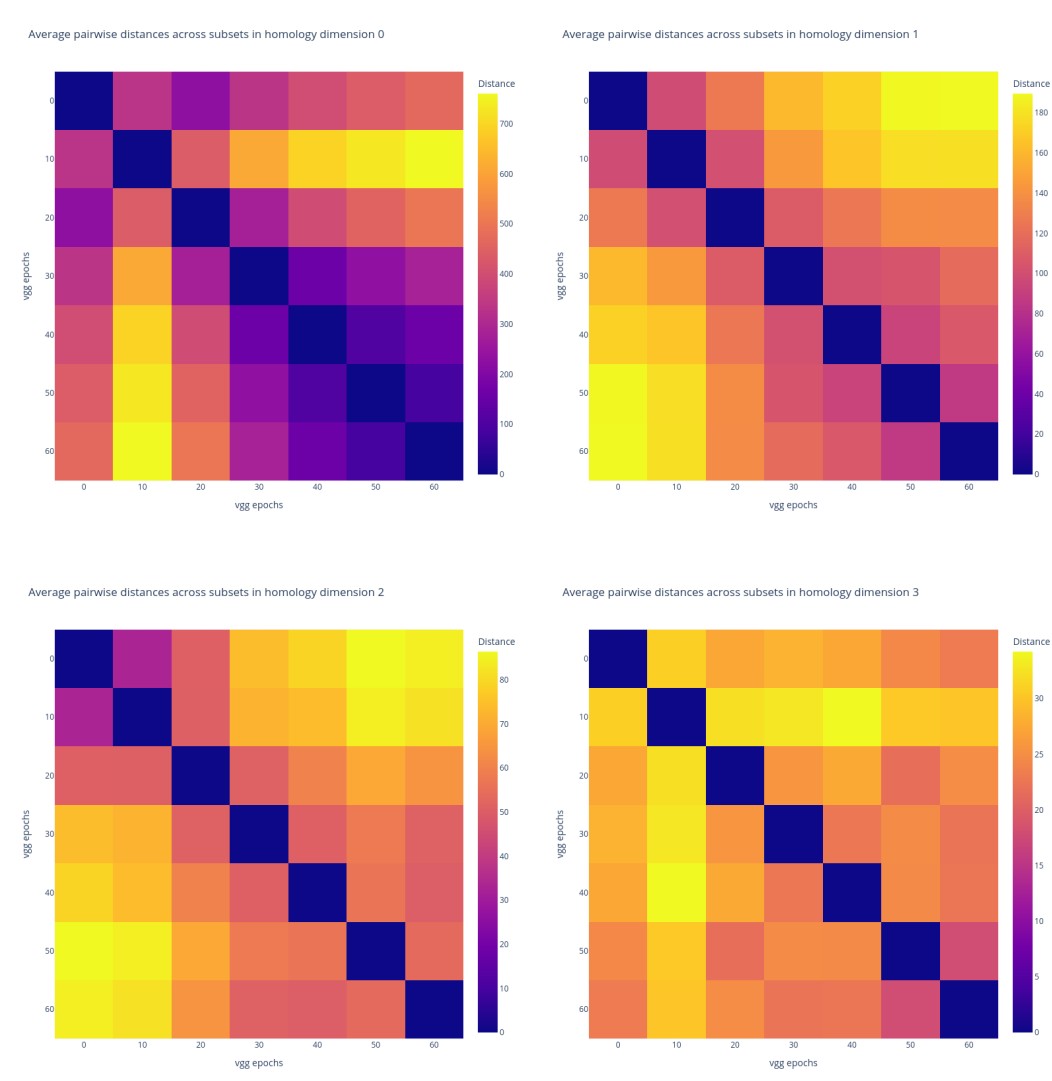

Figure 12: Average similarities over all subsets of VGG-16 for each persistent homology dimension.

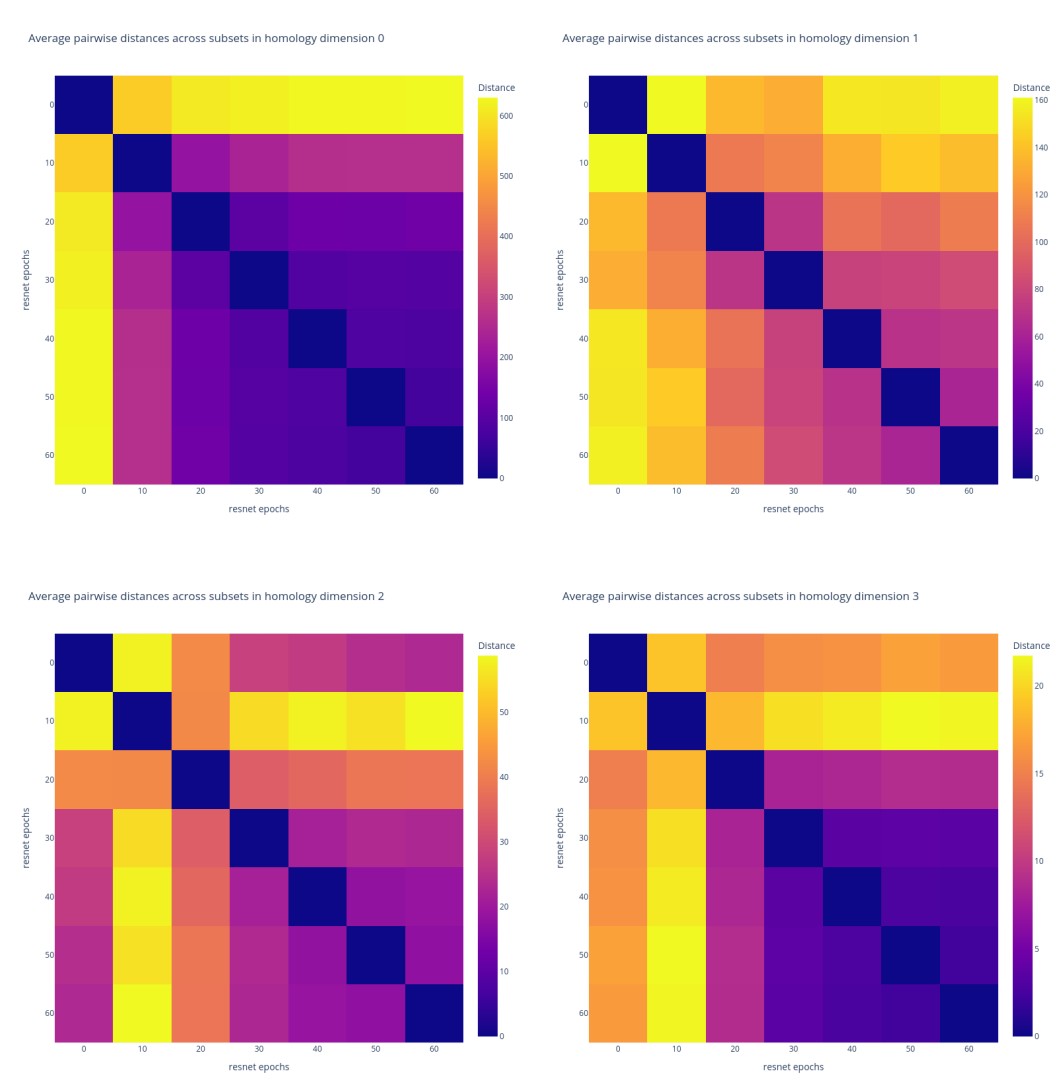

Figure 13: Average similarities over all subsets of ResNet-18 for each persistent homology dimension.

### A.1.2 SELF SIMILARITY ACROSS DATA

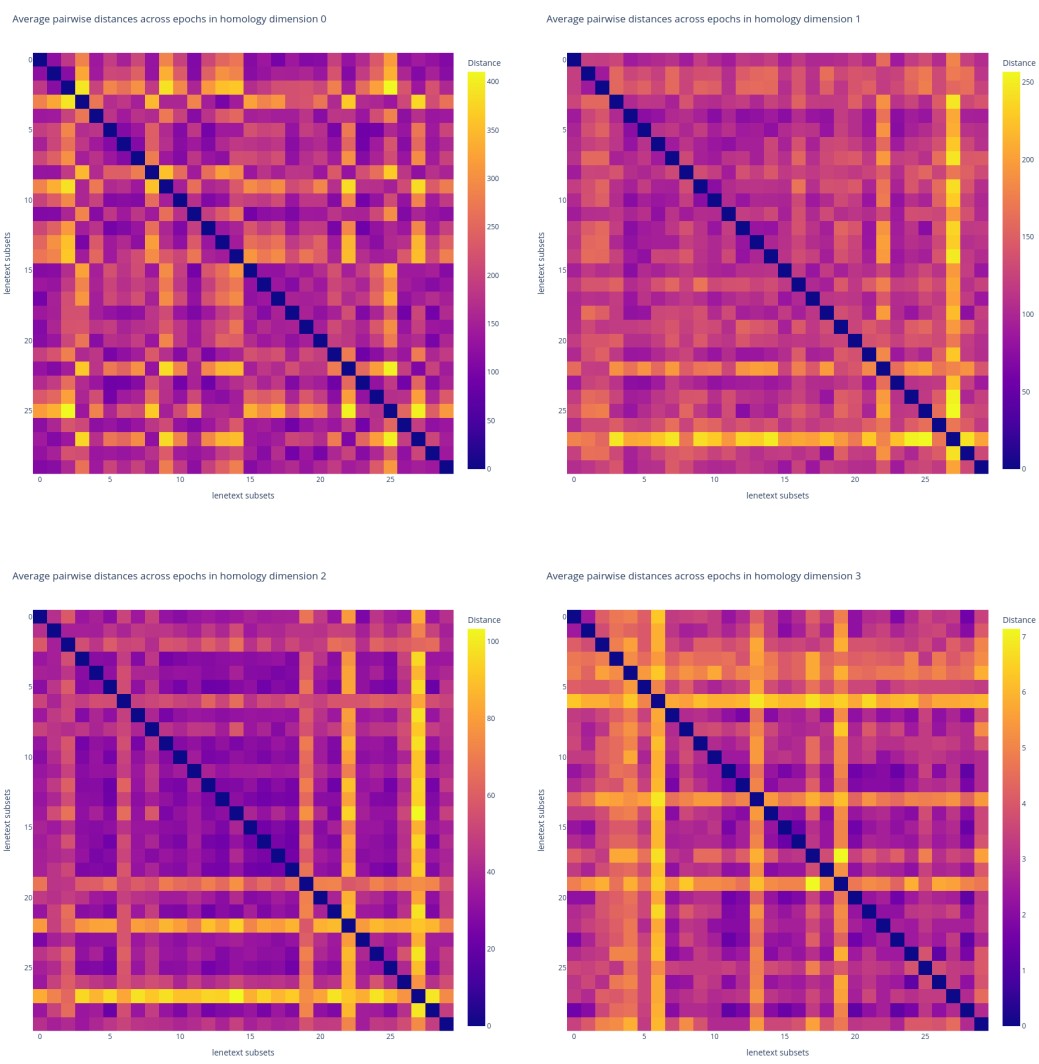

Figure 14: Average similarities over all epochs of LenetExt for each persistent homology dimension.

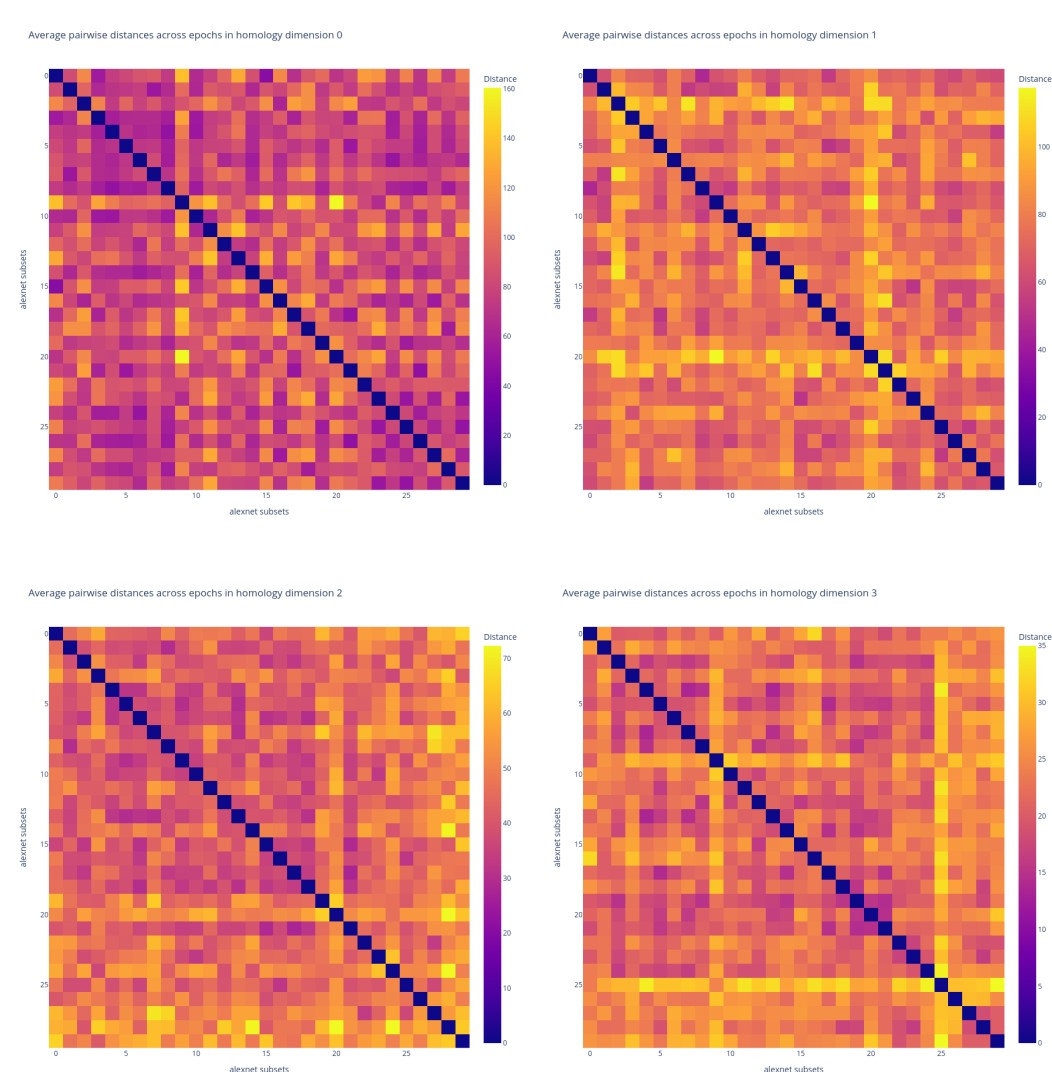

Figure 15: Average similarities over all epochs of AlexNet for each persistent homology dimension.

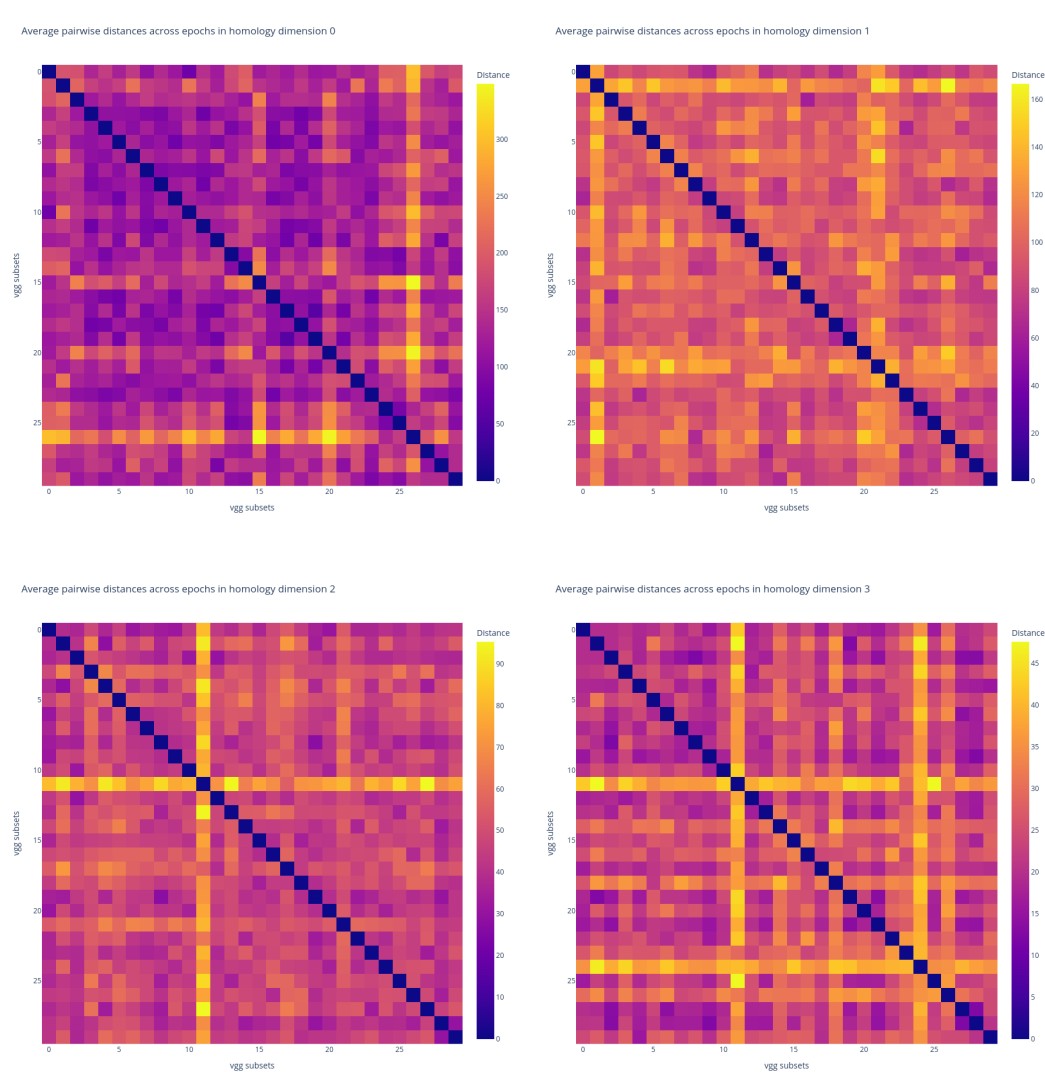

Figure 16: Average similarities over all epochs of VGG-16 for each persistent homology dimension.

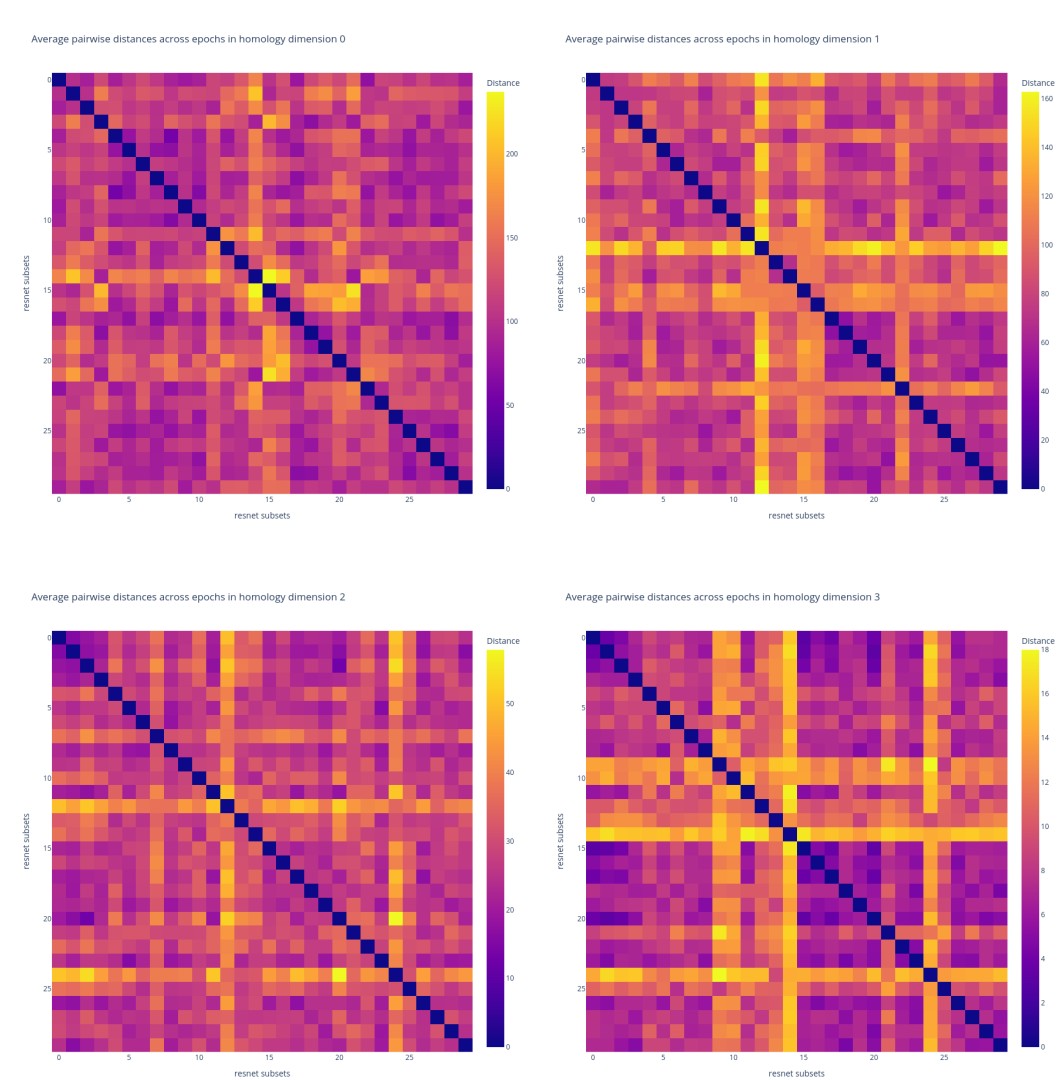

Figure 17: Average similarities over all epochs of ResNet-18 for each persistent homology dimension.

### A.1.3 SIMILARITY ACROSS MODELS AND TIME

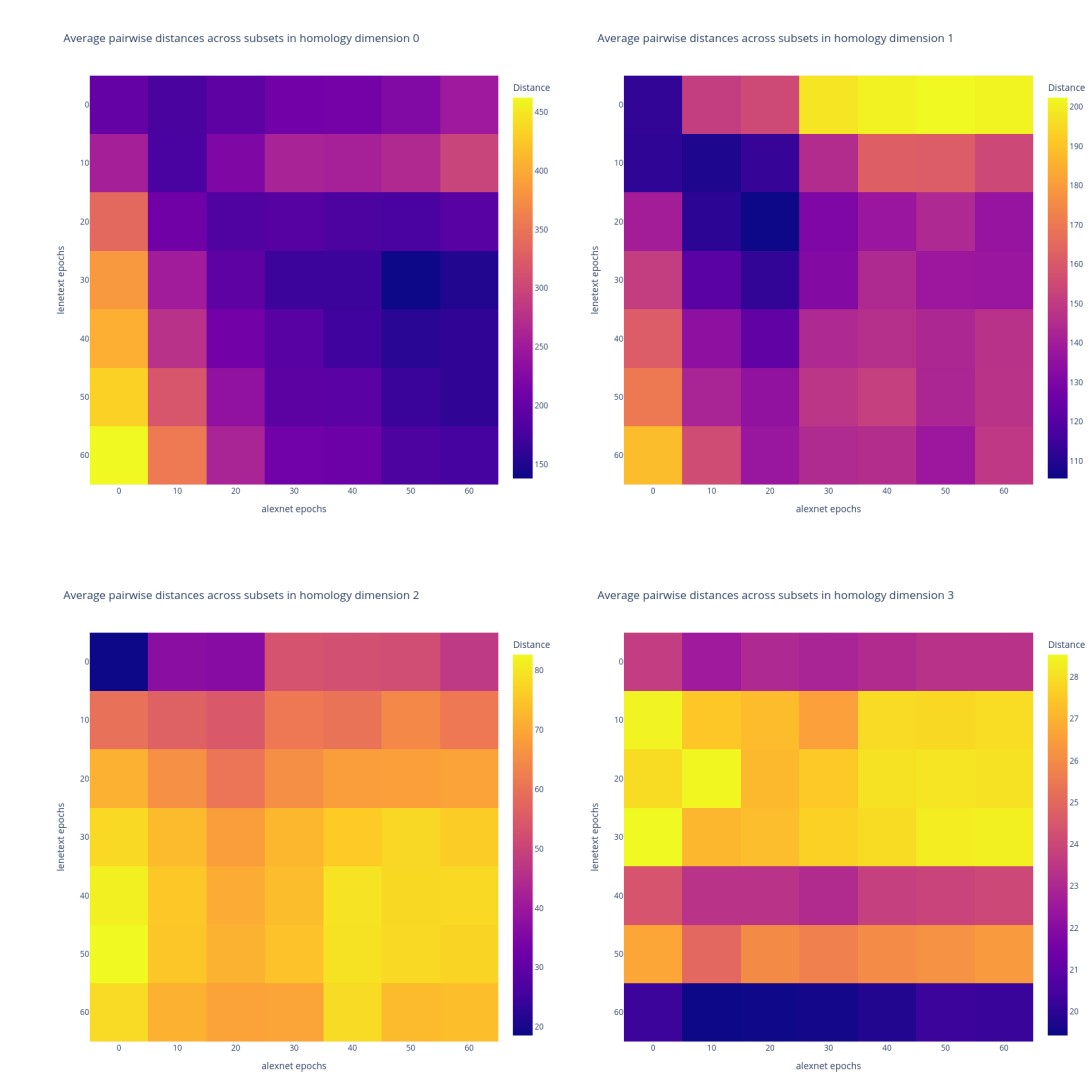

Figure 18: Average similarities over all subsets of LenetExt compared to AlexNet for each persistent homology dimension.

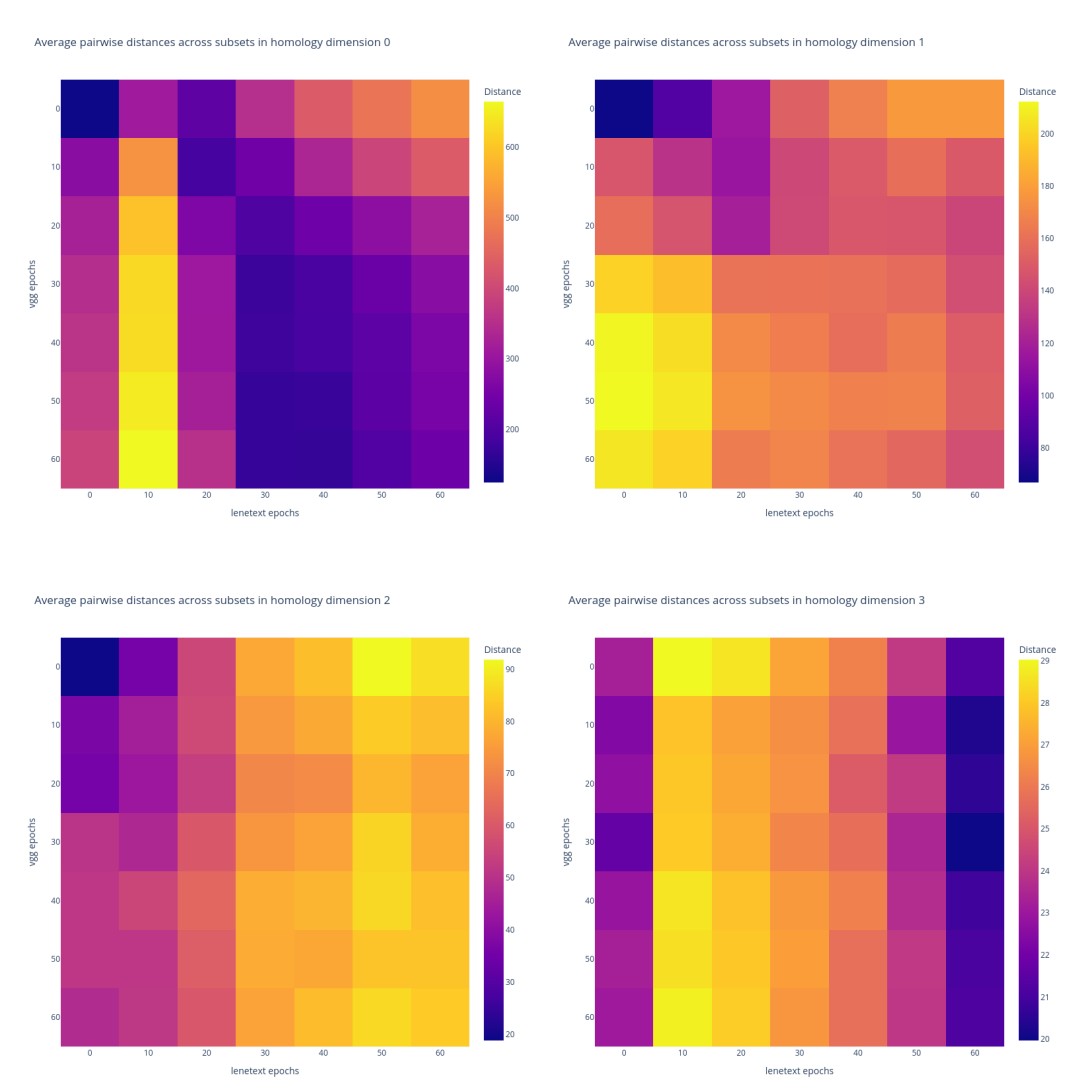

Figure 19: Average similarities over all subsets of LenetExt compared to VGG-16 for each persistent homology dimension.

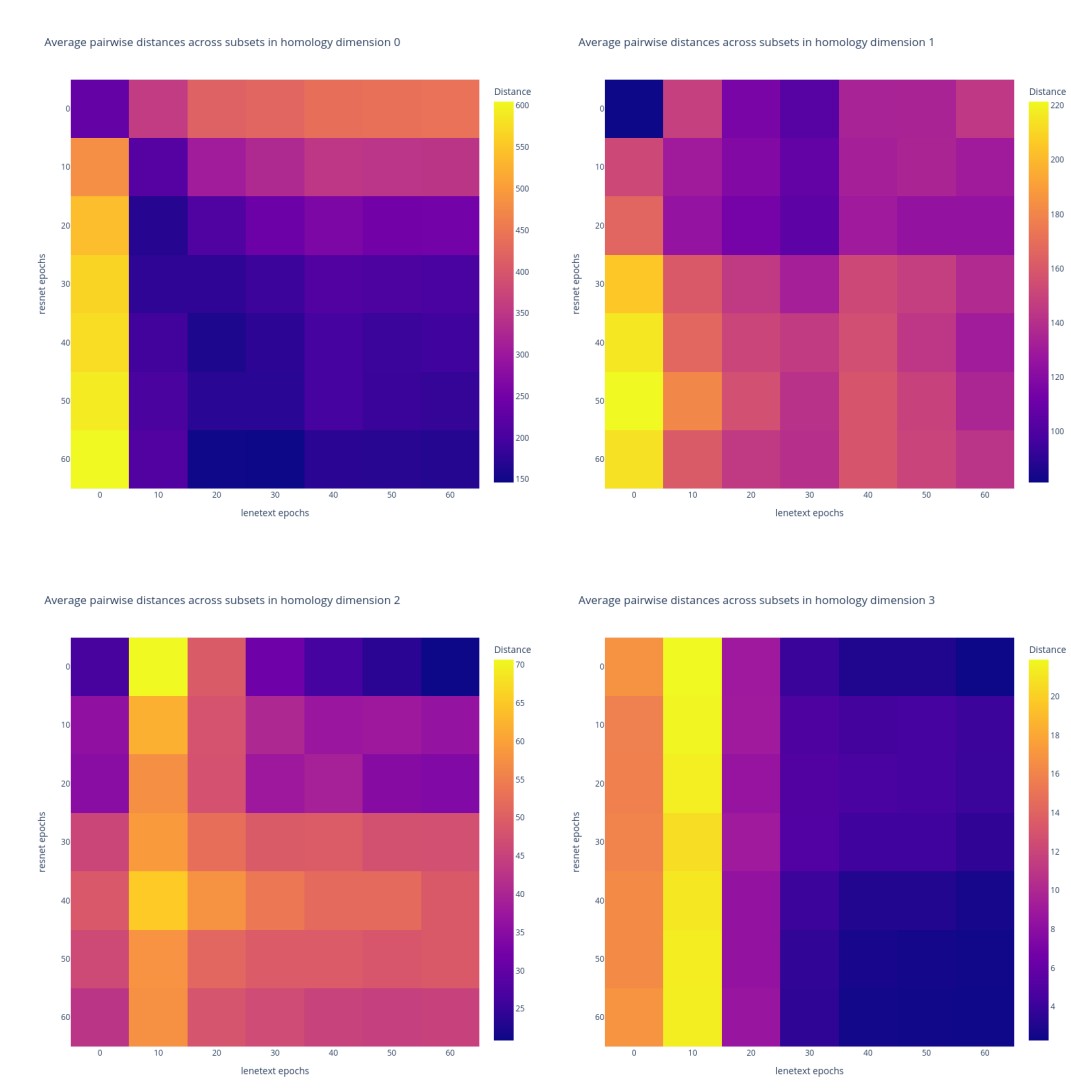

Figure 20: Average similarities over all subsets of LenetExt compared to ResNet for each persistent homology dimension.

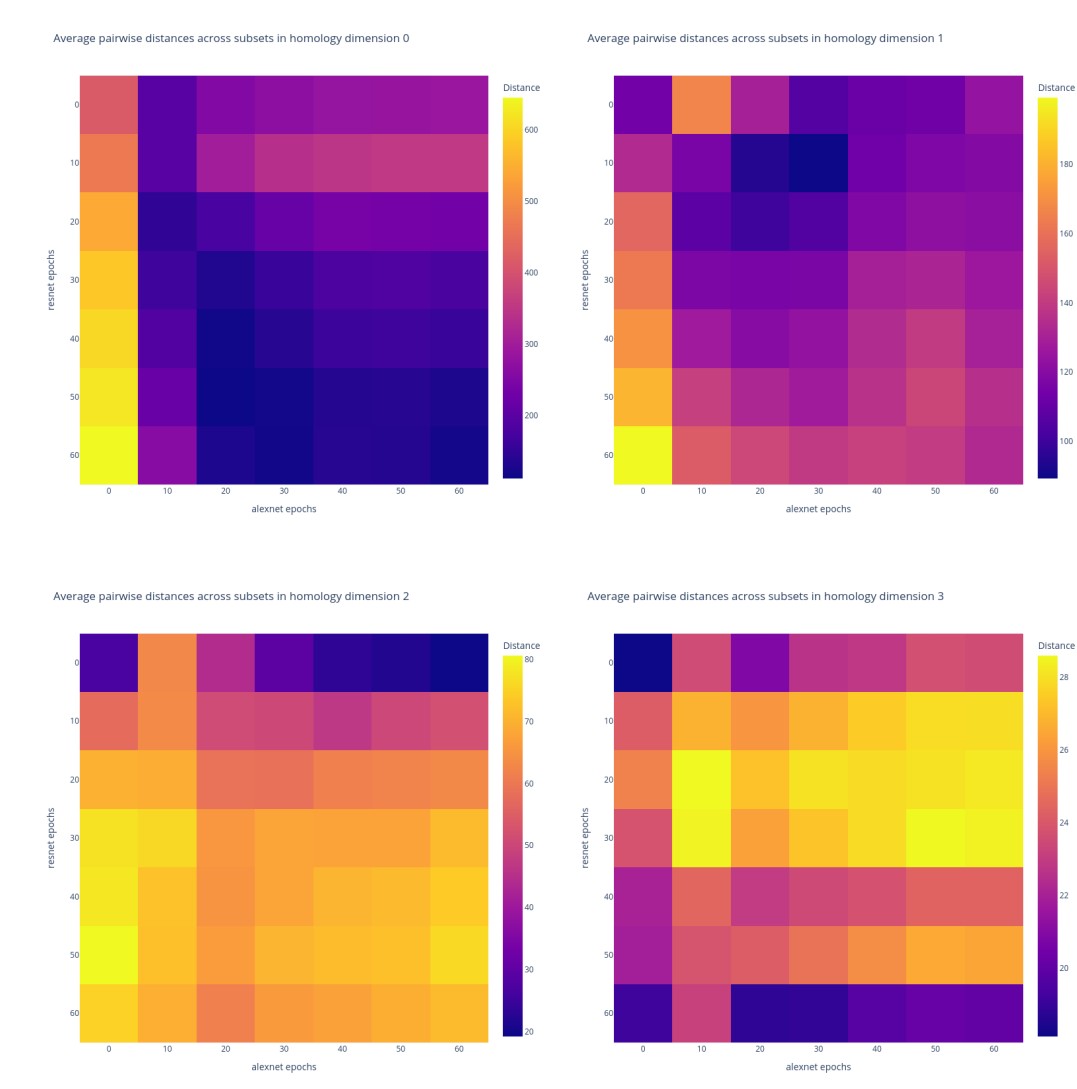

Figure 21: Average similarities over all subsets of AlexNet compared to ResNet-18 for each persistent homology dimension.

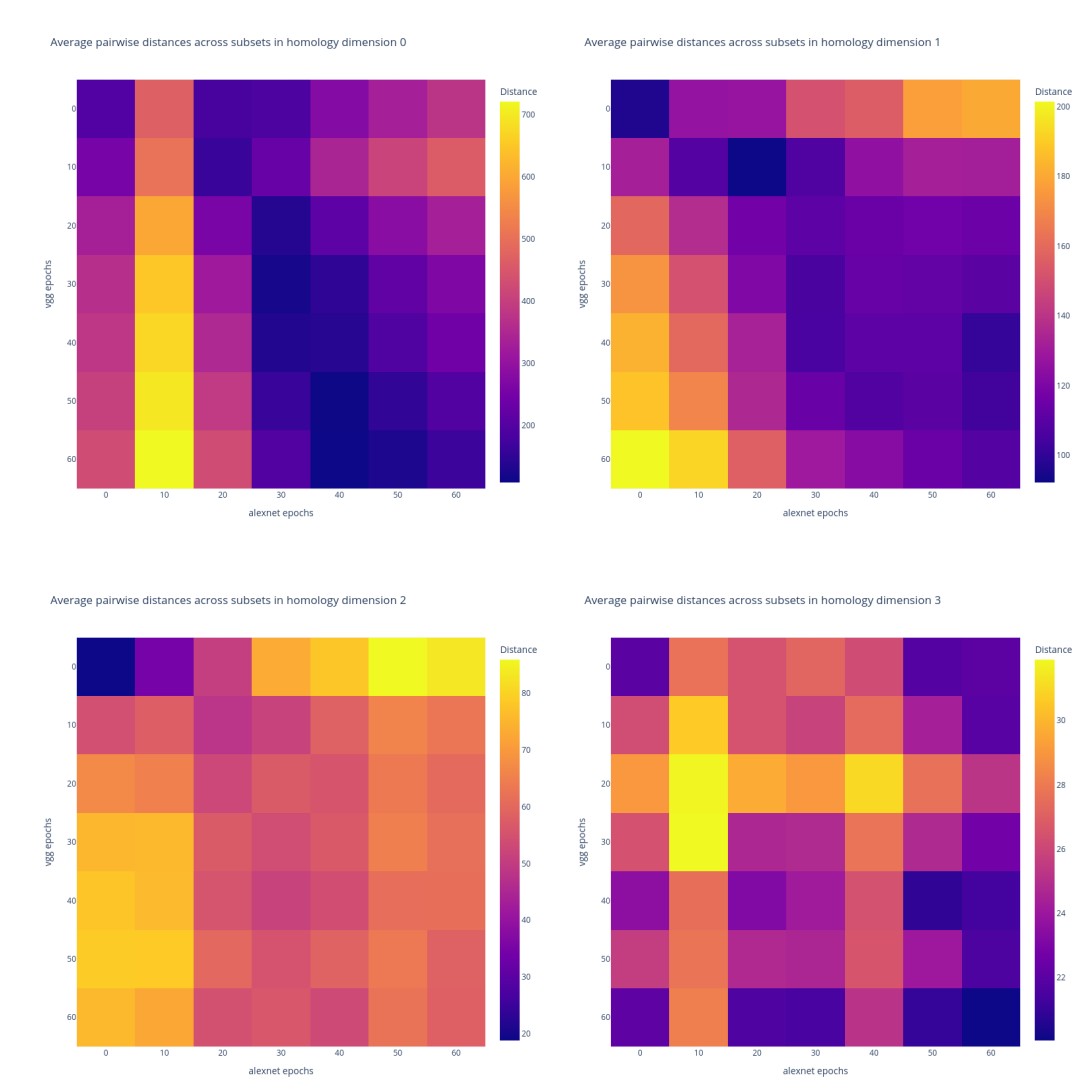

Figure 22: Average similarities over all subsets of AlexNet compared to VGG-16 for each persistent homology dimension.

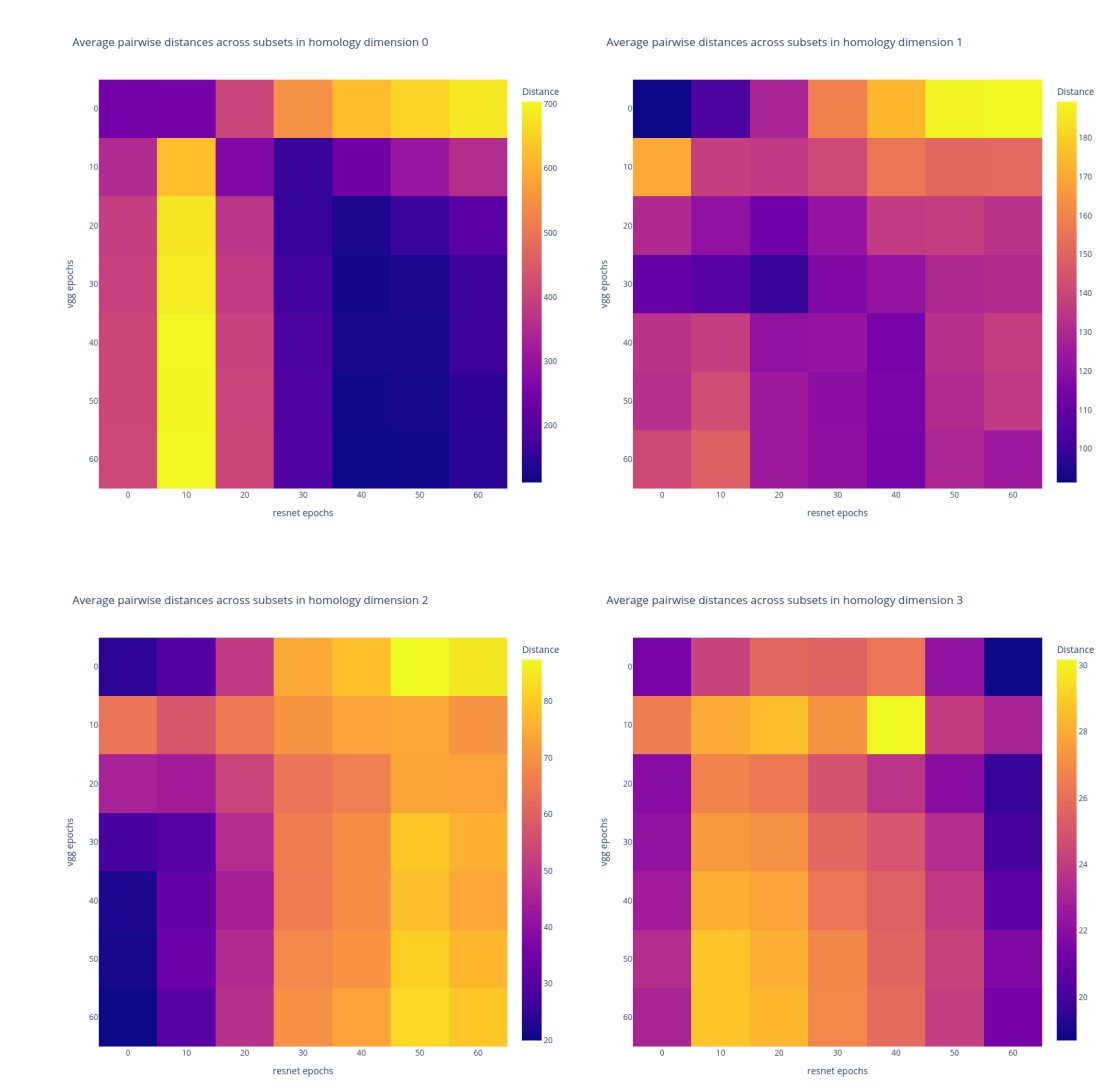

Figure 23: Average similarities over all subsets of VGG-16 compared to ResNet-18 for each persistent homology dimension.

### A.1.4 SIMILARITY ACROSS MODELS AND DATA

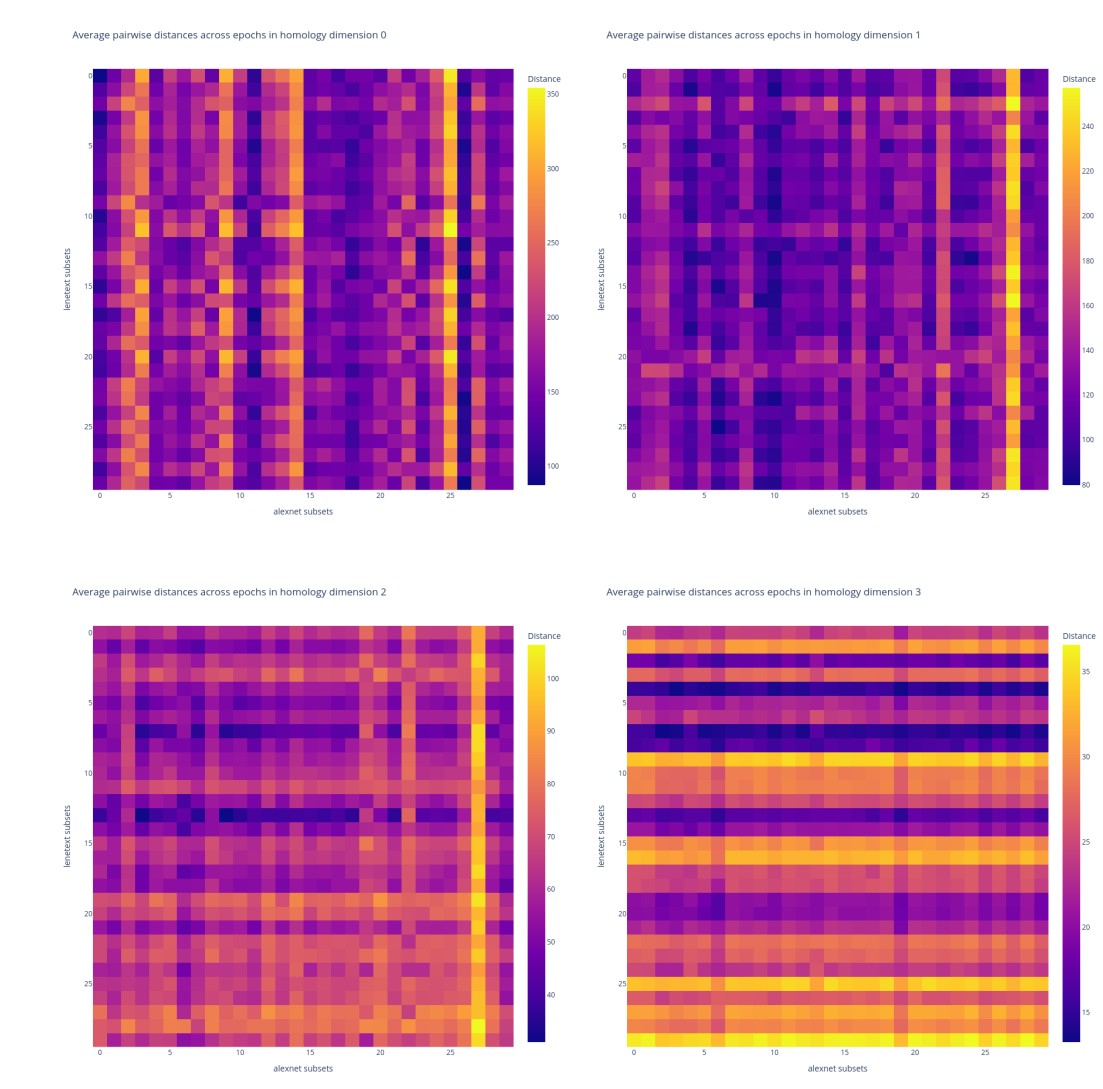

Figure 24: Average similarities over all epochs of LenetExt compared to AlexNet for each persistent homology dimension.

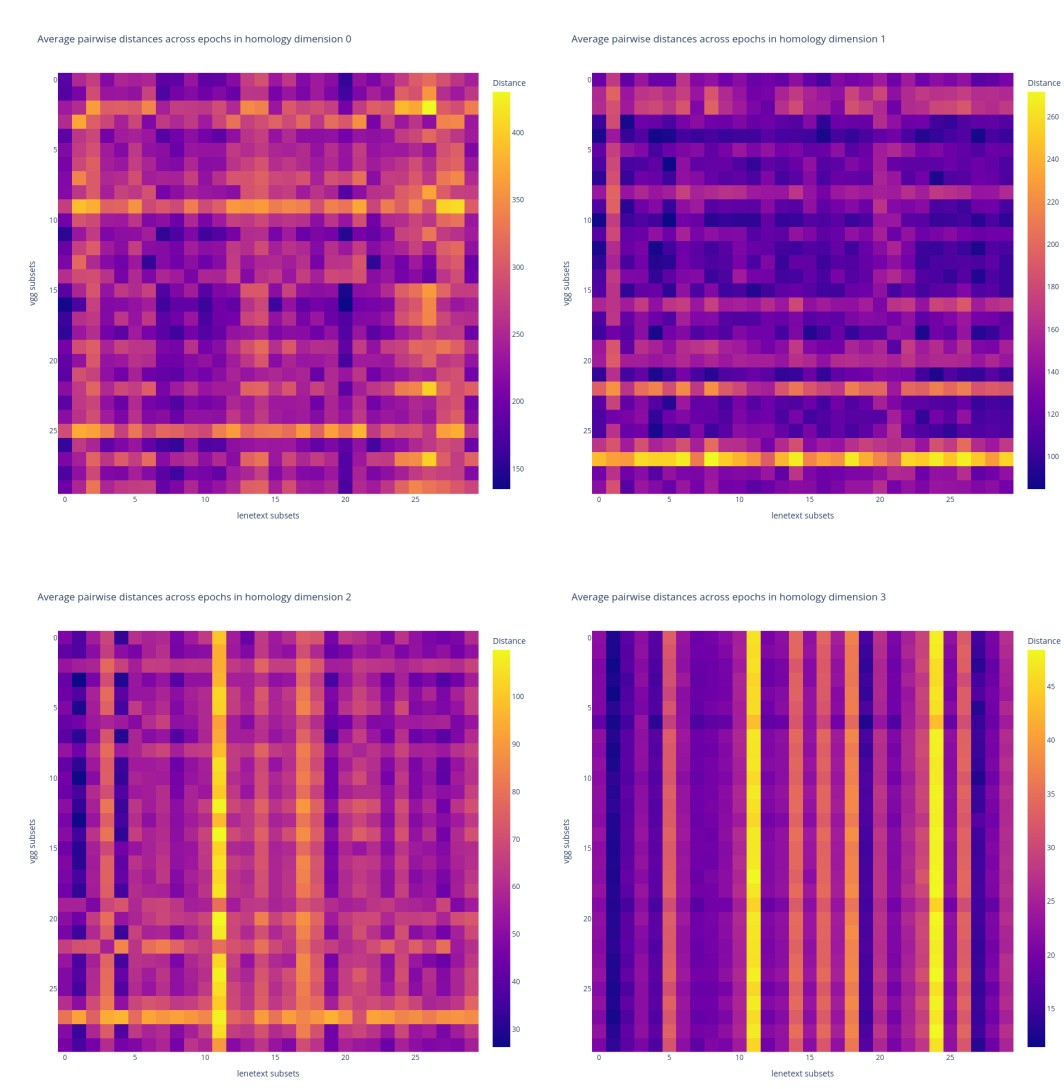

Figure 25: Average similarities over all epochs of LenetExt compared to VGG-16 for each persistent homology dimension.

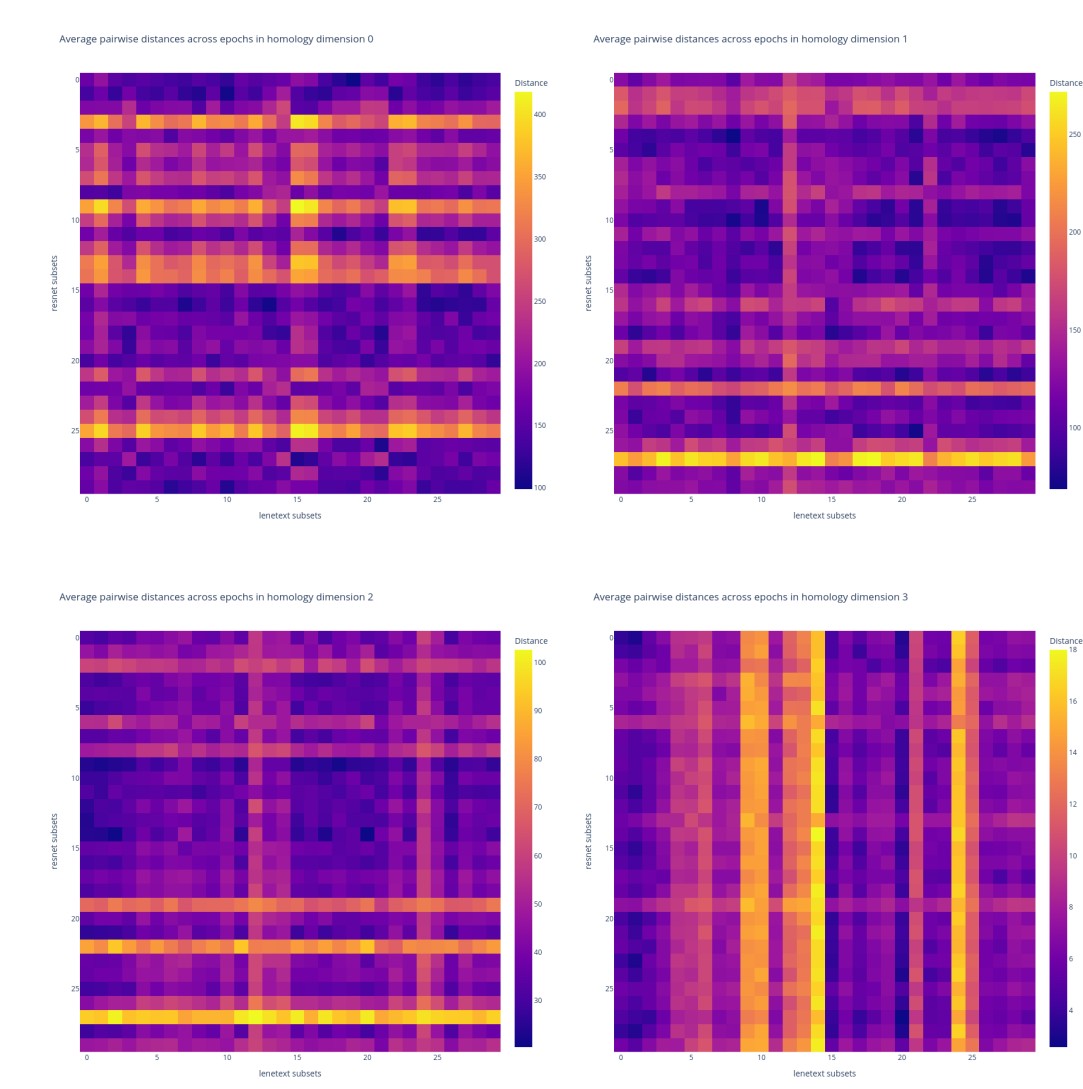

Figure 26: Average similarities over all epochs of LenetExt compared to ResNet for each persistent homology dimension.

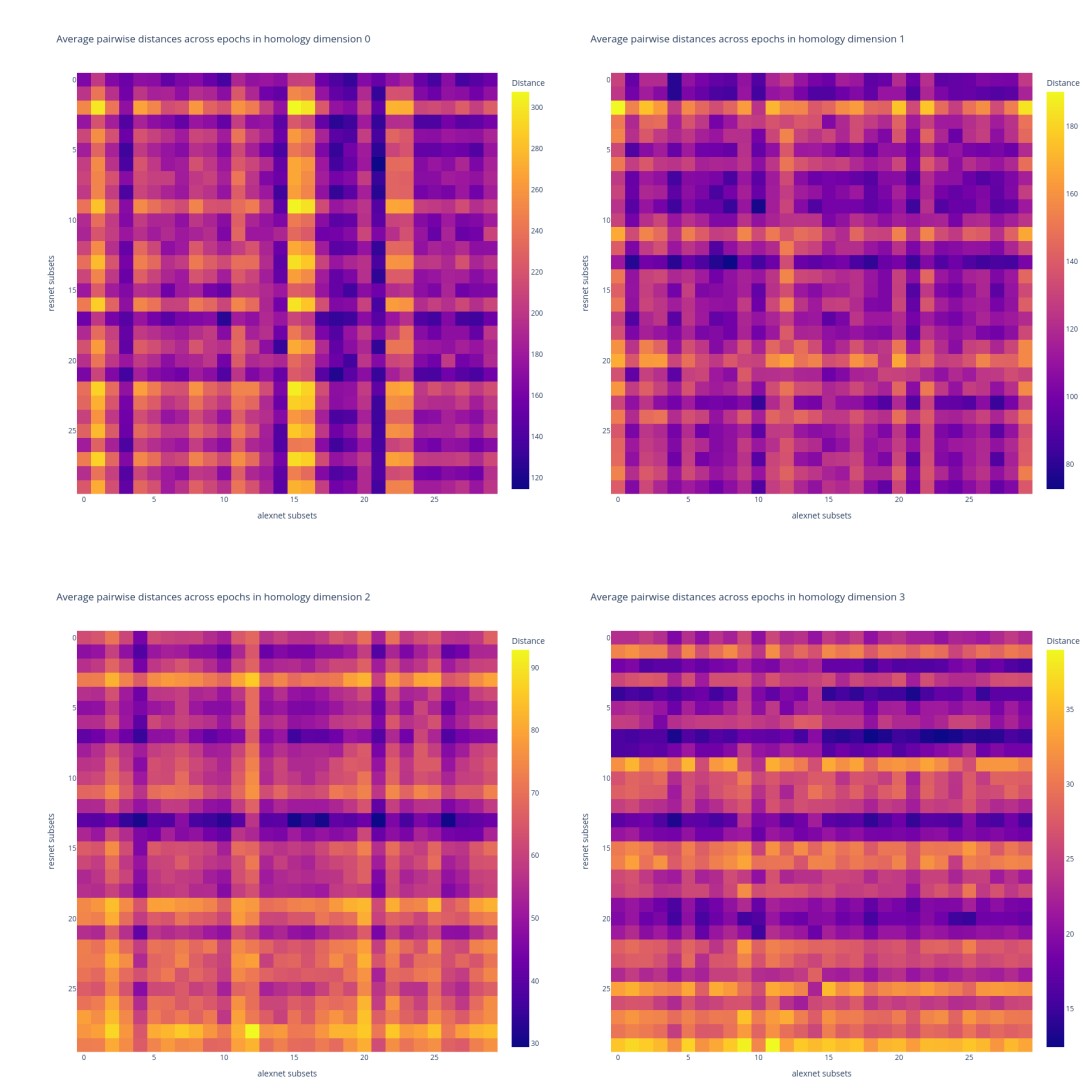

Figure 27: Average similarities over all epochs of AlexNet compared to ResNet-18 for each persistent homology dimension.

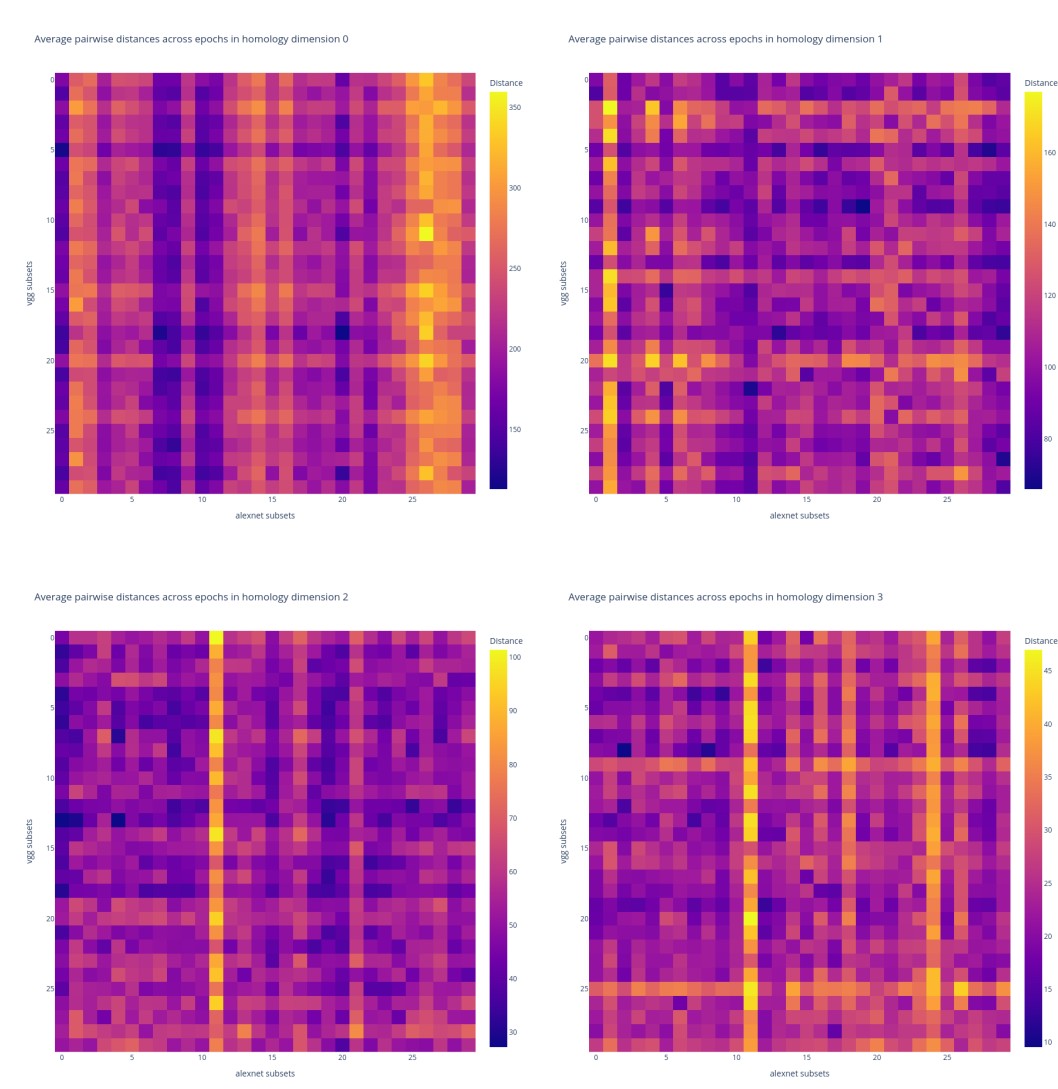

Figure 28: Average similarities over all epochs of AlexNet compared to VGG-16 for each persistent homology dimension.

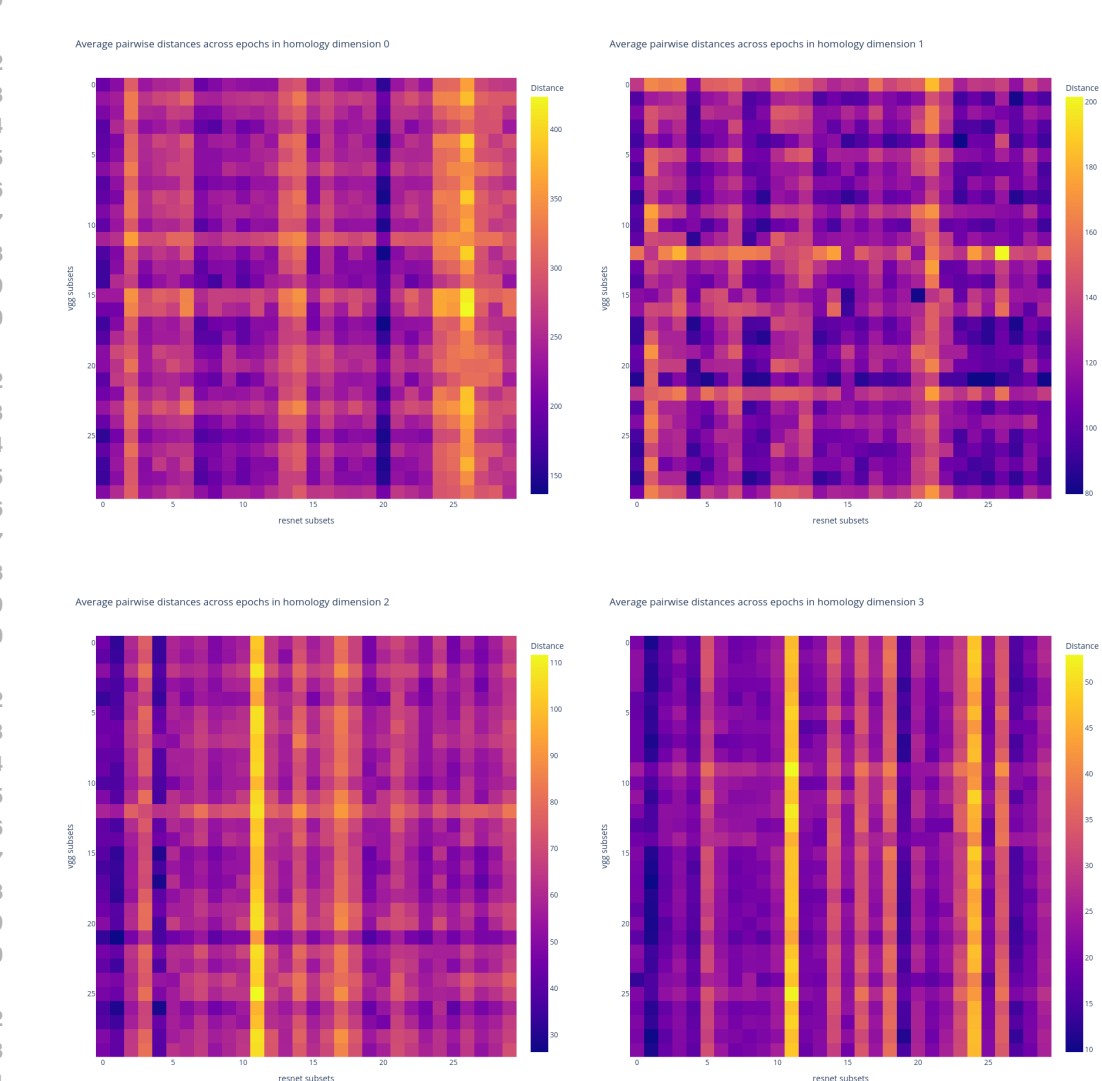

Figure 29: Average similarities over all epochs of VGG-16 compared to ResNet-18 for each persistent homology dimension.

### A.1.5 IMAGENET CLASS LABEL MAPPING

The following is the mapping from the original labels of the ImageNet dataset to the labels used in each of the 30 subsets. The mapping was done in order to ensure that the subsets were disjoint and that the models were trained on different subsets of the dataset. The mapping is as follows:

```
Subset number: 0
Label mapping: n02489166 --> 0 proboscis_monkey
Label mapping: n02097209 --> 1 standard_schnauzer
Label mapping: n09421951 --> 2 sandbar
Label mapping: n02051845 --> 3 pelican
Label mapping: n04004767 --> 4 printer
Label mapping: n02165105 --> 5 tiger_beetle
Label mapping: n04532670 --> 6 viaduct
Label mapping: n02859443 --> 7 boathouse
Label mapping: n03998194 --> 8 prayer_rug
Label mapping: n02815834 --> 9 beaker
```

```
Original subset labels: [682, 991, 156, 769, 439, 364, 689, 556, 621, 100]

Subset number: 1
Label mapping: n02128925 --> 0 jaguar
Label mapping: n02110341 --> 1 dalmatian
Label mapping: n02100583 --> 2 vizsla
Label mapping: n02099712 --> 3 Labrador_retriever
Label mapping: n02012849 --> 4 crane
Label mapping: n01687978 --> 5 agama
Label mapping: n01631663 --> 6 eft
Label mapping: n03404251 --> 7 fur_coat
Label mapping: n15075141 --> 8 toilet_tissue
Label mapping: n03950228 --> 9 pitcher
Original subset labels: [757, 429, 889, 90, 983, 496, 30, 176, 467, 41]

Subset number: 2
Label mapping: n01843383 --> 0 toucan
Label mapping: n01496331 --> 1 electric_ray
Label mapping: n03467068 --> 2 guillotine
Label mapping: n03425413 --> 3 gas_pump
Label mapping: n02167151 --> 4 ground_beetle
Label mapping: n02939185 --> 5 caldron
Label mapping: n04270147 --> 6 spatula
Label mapping: n06596364 --> 7 comic_book
Label mapping: n03187595 --> 8 dial_telephone
Label mapping: n03729826 --> 9 matchstick
Original subset labels: [567, 517, 930, 984, 959, 445, 673, 676, 623, 417]

Subset number: 3
Label mapping: n02096585 --> 0 Boston_bull
Label mapping: n02097047 --> 1 miniature_schnauzer
Label mapping: n02099429 --> 2 curly-coated_retriever
Label mapping: n04311174 --> 3 steel_drum
Label mapping: n02169497 --> 4 leaf_beetle
Label mapping: n02281787 --> 5 lycaenid
Label mapping: n03920288 --> 6 Petri_dish
Label mapping: n02667093 --> 7 abaya
Label mapping: n06874185 --> 8 traffic_light
Label mapping: n07880968 --> 9 burrito
Original subset labels: [123, 144, 783, 861, 113, 340, 646, 625, 853, 900]

Subset number: 4
Label mapping: n02326432 --> 0 hare
Label mapping: n02091032 --> 1 Italian_greyhound
Label mapping: n03041632 --> 2 cleaver
Label mapping: n01531178 --> 3 goldfinch
Label mapping: n01728920 --> 4 ringneck_snake
Label mapping: n02256656 --> 5 cicada
Label mapping: n02927161 --> 6 butcher_shop
Label mapping: n04443257 --> 7 tobacco_shop
Label mapping: n03291819 --> 8 envelope
Label mapping: n04026417 --> 9 purse
Original subset labels: [189, 939, 129, 879, 636, 707, 370, 478, 710, 387]

Subset number: 5
Label mapping: n02132136 --> 0 brown_bear
Label mapping: n02097298 --> 1 Scotch_terrier
Label mapping: n01514668 --> 2 cock
Label mapping: n01494475 --> 3 hammerhead
```

```
Label mapping: n01751748 --> 4 sea_snake
Label mapping: n04067472 --> 5 reel
Label mapping: n06785654 --> 6 crossword_puzzle
Label mapping: n04350905 --> 7 suit
Label mapping: n04118538 --> 8 rugby_ball
Label mapping: n03916031 --> 9 perfume
Original subset labels: [570, 791, 883, 490, 109, 794, 383, 876, 444, 61]

Subset number: 6
Label mapping: n02108000 --> 0 EntleBucher
Label mapping: n02099601 --> 1 golden_retriever
Label mapping: n02398521 --> 2 hippopotamus
Label mapping: n03394916 --> 3 French_horn
Label mapping: n01807496 --> 4 partridge
Label mapping: n01704323 --> 5 triceratops
Label mapping: n04044716 --> 6 radio_telescope
Label mapping: n04019541 --> 7 puck
Label mapping: n07718472 --> 8 cucumber
Label mapping: n07836838 --> 9 chocolate_sauce
Original subset labels: [953, 348, 572, 474, 407, 125, 537, 743, 167, 79]

Subset number: 7
Label mapping: n02488291 --> 0 langur
Label mapping: n04507155 --> 1 umbrella
Label mapping: n02930766 --> 2 cab
Label mapping: n03770679 --> 3 minivan
Label mapping: n02992211 --> 4 cello
Label mapping: n01641577 --> 5 bullfrog
Label mapping: n04127249 --> 6 safety_pin
Label mapping: n01773157 --> 7 black_and_gold_garden_spider
Label mapping: n03877472 --> 8 pajama
Label mapping: n03938244 --> 9 pillow
Original subset labels: [759, 888, 604, 267, 342, 586, 499, 220, 271, 203]

Subset number: 8
Label mapping: n02089973 --> 0 English_foxhound
Label mapping: n04483307 --> 1 trimaran
Label mapping: n01688243 --> 2 frilled_lizard
Label mapping: n04579432 --> 3 whistle
Label mapping: n02871525 --> 4 bookshop
Label mapping: n04493381 --> 5 tub
Label mapping: n04476259 --> 6 tray
Label mapping: n02877765 --> 7 bottlecap
Label mapping: n02869837 --> 8 bonnet
Label mapping: n03240683 --> 9 drilling_platform
Original subset labels: [766, 468, 779, 207, 706, 242, 805, 834, 765, 502]

Subset number: 9
Label mapping: n02071294 --> 0 killer_whale
Label mapping: n02093647 --> 1 Bedlington_terrier
Label mapping: n02074367 --> 2 dugong
Label mapping: n04252077 --> 3 snowmobile
Label mapping: n07749582 --> 4 lemon
Label mapping: n09472597 --> 5 volcano
Label mapping: n04592741 --> 6 wing
Label mapping: n02963159 --> 7 cardigan
Label mapping: n02669723 --> 8 academic_gown
Label mapping: n06794110 --> 9 street_sign
Original subset labels: [362, 503, 320, 22, 288, 896, 193, 836, 932, 119]
```

```
Subset number: 10
Label mapping: n02492035 --> 0 capuchin
Label mapping: n02395406 --> 1 hog
Label mapping: n02130308 --> 2 cheetah
Label mapping: n07745940 --> 3 strawberry
Label mapping: n02687172 --> 4 aircraft_carrier
Label mapping: n04465501 --> 5 tractor
Label mapping: n03649909 --> 6 lawn_mower
Label mapping: n01749939 --> 7 green_mamba
Label mapping: n04548362 --> 8 wallet
Label mapping: n03680355 --> 9 Loafer
Original subset labels: [142, 246, 206, 229, 147, 928, 973, 289, 374, 489]

Subset number: 11
Label mapping: n02101556 --> 0 clumber
Label mapping: n01484850 --> 1 great_white_shark
Label mapping: n03876231 --> 2 paintbrush
Label mapping: n03208938 --> 3 disk_brake
Label mapping: n01784675 --> 4 centipede
Label mapping: n04229816 --> 5 ski_mask
Label mapping: n04357314 --> 6 sunscreen
Label mapping: n04487081 --> 7 trolleybus
Label mapping: n02978881 --> 8 cassette
Label mapping: n03710193 --> 9 mailbox
Original subset labels: [94, 882, 579, 611, 504, 810, 917, 776, 890, 442]

Subset number: 12
Label mapping: n02123394 --> 0 Persian_cat
Label mapping: n02123597 --> 1 Siamese_cat
Label mapping: n03131574 --> 2 crib
Label mapping: n04344873 --> 3 studio_couch
Label mapping: n03075370 --> 4 combination_lock
Label mapping: n03803284 --> 5 muzzle
Label mapping: n03207941 --> 6 dishwasher
Label mapping: n02817516 --> 7 bearskin
Label mapping: n03782006 --> 8 monitor
Label mapping: n04235860 --> 9 sleeping_bag
Original subset labels: [667, 95, 849, 943, 311, 583, 298, 588, 869, 10]

Subset number: 13
Label mapping: n02087394 --> 0 Rhodesian_ridgeback
Label mapping: n12057211 --> 1 yellow_lady's_slipper
Label mapping: n02526121 --> 2 eel
Label mapping: n01742172 --> 3 boa_constrictor
Label mapping: n04355338 --> 4 sundial
Label mapping: n02879718 --> 5 bow
Label mapping: n03787032 --> 6 mortarboard
Label mapping: n02786058 --> 7 Band_Aid
Label mapping: n03584254 --> 8 iPod
Label mapping: n03063599 --> 9 coffee_mug
Original subset labels: [967, 854, 538, 451, 486, 996, 200, 358, 526, 980]

Subset number: 14
Label mapping: n02104365 --> 0 schipperke
Label mapping: n02093991 --> 1 Irish_terrier
Label mapping: n02487347 --> 2 macaque
Label mapping: n02109961 --> 3 Eskimo_dog
Label mapping: n02088238 --> 4 basset
```

```
Label mapping: n04252225 --> 5 snowplow
Label mapping: n12144580 --> 6 corn
Label mapping: n03109150 --> 7 corkscrew
Label mapping: n04153751 --> 8 screw
Label mapping: n03657121 --> 9 lens_cap
Original subset labels: [587, 161, 331, 126, 278, 988, 68, 376, 138, 149]

Subset number: 15
Label mapping: n02415577 --> 0 bighorn
Label mapping: n02342885 --> 1 hamster
Label mapping: n03478589 --> 2 half_track
Label mapping: n02643566 --> 3 lionfish
Label mapping: n01669191 --> 4 box_turtle
Label mapping: n02699494 --> 5 altar
Label mapping: n03062245 --> 6 cocktail_shaker
Label mapping: n03617480 --> 7 kimono
Label mapping: n12985857 --> 8 coral_fungus
Label mapping: n03188531 --> 9 diaper
Original subset labels: [761, 249, 52, 886, 770, 157, 677, 454, 966, 462]

Subset number: 16
Label mapping: n02484975 --> 0 guenon
Label mapping: n02090622 --> 1 borzoi
Label mapping: n02095314 --> 2 wire-haired_fox_terrier
Label mapping: n01872401 --> 3 echidna
Label mapping: n03452741 --> 4 grand_piano
Label mapping: n12267677 --> 5 acorn
Label mapping: n03627232 --> 6 knot
Label mapping: n07716906 --> 7 spaghetti_squash
Label mapping: n07932039 --> 8 eggnog
Label mapping: n04553703 --> 9 washbasin
Original subset labels: [215, 105, 73, 582, 327, 740, 227, 906, 160, 823]

Subset number: 17
Label mapping: n03344393 --> 0 fireboat
Label mapping: n03100240 --> 1 convertible
Label mapping: n03742115 --> 2 medicine_chest
Label mapping: n02676566 --> 3 acoustic_guitar
Label mapping: n09468604 --> 4 valley
Label mapping: n01537544 --> 5 indigo_bunting
Label mapping: n04330267 --> 6 stove
Label mapping: n03042490 --> 7 cliff_dwelling
Label mapping: n03000134 --> 8 chainlink_fence
Label mapping: n13054560 --> 9 bolete
Original subset labels: [721, 516, 390, 268, 981, 235, 713, 302, 345, 360]

Subset number: 18
Label mapping: n02423022 --> 0 gazelle
Label mapping: n04310018 --> 1 steam_locomotive
Label mapping: n04467665 --> 2 trailer_truck
Label mapping: n04429376 --> 3 throne
Label mapping: n03290653 --> 4 entertainment_center
Label mapping: n01806567 --> 5 quail
Label mapping: n02980441 --> 6 castle
Label mapping: n02791270 --> 7 barbershop
Label mapping: n04296562 --> 8 stage
Label mapping: n04033901 --> 9 quill
Original subset labels: [804, 283, 12, 701, 406, 308, 263, 705, 316, 862]
```

```
Subset number: 19
Label mapping: n02091831 --> 0 Saluki
Label mapping: n02110806 --> 1 basenji
Label mapping: n02108551 --> 2 Tibetan_mastiff
Label mapping: n04552348 --> 3 warplane
Label mapping: n07753113 --> 4 fig
Label mapping: n02951585 --> 5 can_opener
Label mapping: n01796340 --> 6 ptarmigan
Label mapping: n03483316 --> 7 hand_blower
Label mapping: n03814639 --> 8 neck_brace
Label mapping: n03903868 --> 9 pedestal
Original subset labels: [231, 84, 505, 110, 321, 377, 732, 595, 66, 402]

Subset number: 20
Label mapping: n02124075 --> 0 Egyptian_cat
Label mapping: n02086910 --> 1 papillon
Label mapping: n02091467 --> 2 Norwegian_elkhound
Label mapping: n03393912 --> 3 freight_car
Label mapping: n03777568 --> 4 Model_T
Label mapping: n04461696 --> 5 tow_truck
Label mapping: n04065272 --> 6 recreational_vehicle
Label mapping: n01592084 --> 7 chickadee
Label mapping: n04023962 --> 8 punching_bag
Label mapping: n02783161 --> 9 ballpoint
Original subset labels: [282, 43, 256, 907, 395, 8, 272, 63, 286, 846]

Subset number: 21
Label mapping: n02119022 --> 0 red_fox
Label mapping: n09428293 --> 1 seashore
Label mapping: n04548280 --> 2 wall_clock
Label mapping: n02236044 --> 3 mantis
Label mapping: n02264363 --> 4 lacewing
Label mapping: n04366367 --> 5 suspension_bridge
Label mapping: n03837869 --> 6 obelisk
Label mapping: n07590611 --> 7 hot_pot
Label mapping: n03388183 --> 8 fountain_pen
Label mapping: n04325704 --> 9 stole
Original subset labels: [635, 62, 699, 998, 367, 681, 934, 524, 771, 638]

Subset number: 22
Label mapping: n02088364 --> 0 beagle
Label mapping: n02093428 --> 1 American_Staffordshire_terrier
Label mapping: n03642806 --> 2 laptop
Label mapping: n04037443 --> 3 racer
Label mapping: n01829413 --> 4 hornbill
Label mapping: n02006656 --> 5 spoonbill
Label mapping: n02892201 --> 6 brass
Label mapping: n02730930 --> 7 apron
Label mapping: n02808440 --> 8 bathtub
Label mapping: n03866082 --> 9 overskirt
Original subset labels: [273, 884, 716, 228, 414, 937, 132, 845, 170, 424]

Subset number: 23
Label mapping: n02092339 --> 0 Weimaraner
Label mapping: n02672831 --> 1 accordion
Label mapping: n04482393 --> 2 tricycle
Label mapping: n04154565 --> 3 screwdriver
Label mapping: n01532829 --> 4 house_finch
Label mapping: n01601694 --> 5 water_ouzel
```

```
Label mapping: n01756291 --> 6 sidewinder
Label mapping: n03841143 --> 7 odometer
Label mapping: n04418357 --> 8 theater_curtain
Label mapping: n02802426 --> 9 basketball
Original subset labels: [493, 388, 908, 291, 903, 379, 520, 396, 25, 223]

Subset number: 24
Label mapping: n02443484 --> 0 black-footed_ferret
Label mapping: n02109525 --> 1 Saint_Bernard
Label mapping: n02105251 --> 2 briard
Label mapping: n07753592 --> 3 banana
Label mapping: n01582220 --> 4 magpie
Label mapping: n02002724 --> 5 black_stork
Label mapping: n02011460 --> 6 bittern
Label mapping: n01734418 --> 7 king_snake
Label mapping: n02165456 --> 8 ladybug
Label mapping: n07693725 --> 9 bagel
Original subset labels: [208, 323, 177, 40, 622, 428, 423, 768, 481, 394]

Subset number: 25
Label mapping: n02094433 --> 0 Yorkshire_terrier
Label mapping: n02110627 --> 1 affenpinscher
Label mapping: n02100236 --> 2 German_short-haired_pointer
Label mapping: n02328150 --> 3 Angora
Label mapping: n02804610 --> 4 bassoon
Label mapping: n01498041 --> 5 stingray
Label mapping: n01985128 --> 6 crayfish
Label mapping: n01944390 --> 7 snail
Label mapping: n04277352 --> 8 spindle
Label mapping: n09835506 --> 9 ballplayer
Original subset labels: [653, 353, 619, 954, 127, 59, 775, 446, 164, 134]

Subset number: 26
Label mapping: n02112706 --> 0 Brabancon_griffon
Label mapping: n02364673 --> 1 guinea_pig
Label mapping: n01616318 --> 2 vulture
Label mapping: n01740131 --> 3 night_snake
Label mapping: n01644900 --> 4 tailed_frog
Label mapping: n01773797 --> 5 garden_spider
Label mapping: n02177972 --> 6 weevil
Label mapping: n04111531 --> 7 rotisserie
Label mapping: n04209239 --> 8 shower_curtain
Label mapping: n02825657 --> 9 bell_cote
Original subset labels: [606, 70, 101, 501, 485, 399, 747, 663, 628, 933]

Subset number: 27
Label mapping: n02417914 --> 0 ibex
Label mapping: n02441942 --> 1 weasel
Label mapping: n02120505 --> 2 grey_fox
Label mapping: n02112350 --> 3 keeshond
Label mapping: n02129165 --> 4 lion
Label mapping: n02981792 --> 5 catamaran
Label mapping: n07760859 --> 6 custard_apple
Label mapping: n01980166 --> 7 fiddler_crab
Label mapping: n07248320 --> 8 book_jacket
Label mapping: n03347037 --> 9 fire_screen
Original subset labels: [774, 919, 67, 325, 48, 148, 241, 9, 190, 615]

Subset number: 28
```

```
Label mapping: n02363005 --> 0 beaver
Label mapping: n03785016 --> 1 moped
Label mapping: n01833805 --> 2 hummingbird
Label mapping: n04228054 --> 3 ski
Label mapping: n01774750 --> 4 tarantula
Label mapping: n02231487 --> 5 walking_stick
Label mapping: n02319095 --> 6 sea_urchin
Label mapping: n04398044 --> 7 teapot
Label mapping: n03899768 --> 8 patio
Label mapping: n03255030 --> 9 dumbbell
Original subset labels: [277, 633, 1000, 657, 608, 415, 675, 590, 679, 195]

Subset number: 29
Label mapping: n02093256 --> 0 Staffordshire_bullterrier
Label mapping: n02111500 --> 1 Great_Pyrenees
Label mapping: n02325366 --> 2 wood_rabbit
Label mapping: n01873310 --> 3 platypus
Label mapping: n04487394 --> 4 trombone
Label mapping: n01677366 --> 5 common_iguana
Label mapping: n01729977 --> 6 green_snake
Label mapping: n04258138 --> 7 solar_dish
Label mapping: n04239074 --> 8 sliding_door
Label mapping: n07684084 --> 9 French_loaf
Original subset labels: [577, 172, 480, 45, 873, 464, 188, 726, 217, 349]
```

