# OpenReview forum: "An Empirical Study on the Application of TDA to Deep Neural Networks"
_ICLR.cc/2025/Conference — Submitted to ICLR 2025_

### Official Review · Reviewer_4ypy · 2024-10-18

**Soundness:** 3
**Presentation:** 2
**Contribution:** 2
**Rating:** 3
**Confidence:** 4

**Summary:**

In their paper "An empirical study of the application of TDA to deep neural networks", the authors develop TDA-based approach to compare deep-learning representations between different layers, different architectures, different datasets, or different training epochs. The authors measure the distance between the two representations as the distance between the corresponding persistence diagrams (more precisely, between Betti curves of the persistence diagrams). They apply this methodology to several CNNs trained on ImageNet and argue that their measure has reasonable behaviour.

**Strengths:**

An interesting and flexible approach to compare neural representations. Comparisons can be made between different architectures, or between different datasets, etc.

**Weaknesses:**

(1) On the method side, there are no comparisons with other possible / existing methods of comparing representation similarity (not involving TDA), e.g. directly based on the pairwise distance matrix. The authors present their approach but do not compare it with anything else really.

(2) On the application side, the observations that the authors make in section 3 are sensible but mostly rather obvious/intuitive. So this serves as a sanity-check but does not present new empirical findings of sufficiently broad interest for ICLR.

Overall, I think the paper does not rise to the level of general interest expected at ICLR.

**Questions:**

MINOR COMMENTS

* Font size in most figures is too small and figures are impossible to read in the print-out.

* line 149: do you pool all neurons from all layers, or is this done per-layer? Please clarify. I assume that all neurons from all layears are used. What is the value of M for the networks you use in the paper? Please give exact numbers.

---

### Official Review · Reviewer_oXpK · 2024-10-23

**Soundness:** 2
**Presentation:** 2
**Contribution:** 1
**Rating:** 1
**Confidence:** 4

**Summary:**

The authors introduce a method for analyzing the global functional structure of deep neural networks derived from comparing Betti curve similarities obtained through persistent homology. The global functional structure of the networks is obtained by clustering the model’s neuron activations into 1000 clusters using the k-Means++ algorithm. The experiments are performed on ImageNet using 4 CNN models.

**Strengths:**

* Good theoretical background of persistence homology and Betti numbers.

**Weaknesses:**

In my opinion, the current version of the paper doesn’t not meet the acceptance criteria, which I support with the following:
* Paper does not have any related work despite the fact that TDA has been applied to analyze representations of NNs before (see for example Geometry Score: A Method For Comparing Generative Adversarial Networks, Khrulkov et al, ICML 2018, or work done by Barannikov like Representation Topology Divergence: A Method for Comparing Neural Network Representations, ICML 2022)
* The experimental setting and the results are very basic. The paper only considers 10 class classification on ImageNet with 4 different CNN models which is not a relevant or realistic setting. How does the method generalize to more categories, to different tasks, architectures, different input sizes?
* The problem is also not well motivated. In what use cases is your method actually useful in practice?
* No ablation studies are provided. The method first performs clustering of the model’s neuron activations into 1000 clusters using the k-Means++ algorithm. The number of clusters affects both the quality and semantics of cluster centroids as well as the results of the persistent homology. No experiments are done on robustness of the parameter.
* Figures are unreadable and sometimes redundant. For example, Figure 1 doesn’t bring any information.

**Questions:**

See weaknesses above.

---

### Official Review · Reviewer_3v38 · 2024-10-29

**Soundness:** 3
**Presentation:** 3
**Contribution:** 3
**Rating:** 6
**Confidence:** 4

**Summary:**

This paper explores the application of Topological Data Analysis (TDA), particularly persistent homology and Betti curve similarity, to analyze the global structure of deep neural networks (DNNs). Using convolutional neural networks (CNNs) trained on subsets of the ImageNet dataset, the authors assess the models' internal structures over time and across different datasets. The study finds that Betti curve similarity can effectively distinguish between different CNN architectures and datasets, offering a new perspective on DNN analysis that complements traditional accuracy metrics.

**Strengths:**

It has clear explanations for persistent homology, and Betti curves as applied to DNN functional graphs. However, there are some approximations in the k-means++ clustering process, leading to potential approximation errors in the persistent homology calculations, which could affect the robustness of the results.

Presentation is clear and well-organized. Figures have explanations with text. But additional background on TDA could enhance accessibility for readers not familiar with this.

It offers a novel approach that adds a new perspective to neural network analysis. This research has the potential to inspire further studies on TDA applications in neural networks and other machine learning models
Strengths:

Originality: The use of Betti curves to analyze DNNs is innovative, and to the best of my knowledge, it has not been applied in this context before. This approach provides an alternative perspective on understanding DNN structure and training dynamics.

Methodology: The authors use well-defined and rigorous methods for data partitioning, clustering, and persistent homology calculation, enhancing the credibility of the results.

Significance: The findings demonstrate the potential of TDA to uncover meaningful insights about DNNs' functional structure, which could influence future research in model interpretability and theoretical DNN analysis

**Weaknesses:**

1. Approximation in Dimensionality Reduction: The k-means++ clustering approximation might introduce errors in the persistent homology analysis. The authors could discuss these limitations in greater detail.
2.	Accessibility of TDA Concepts: For readers not well-versed in TDA, the paper might be challenging to follow. A more comprehensive introduction to TDA concepts would be beneficial.

**Questions:**

1.	Can the authors clarify the role of approximation errors introduced by k-Means++ and how it might impact the conclusions drawn from the Betti curve similarity results?
2.	Would similar trends in Betti curve similarity be observed across other types of data (e.g., text or sequential data)?
3.	Are there specific architectural modifications within the CNNs that would noticeably impact the persistent homology features?

---

> ### Author Response · Authors · 2024-11-20
> **Answers to Questions**
>
> Thank you for taking the time to review our submission.
>
> 1) At this time we are unable to give approximation errors for the k-means++ reduction since computing the non-approximated topology of the functional graphs is computationally intractable.
>
> 2) We don't currently have information on other data types but our hypothesis is, on average yes. For example, I would guess that most people process the same languages similarly and create similar representation spaces for them.
>
> 3) Thank you, another good question. We are in the process of experimenting further with this method and designing ablation studies that could better answer it.

---

> > ### Comment · Reviewer_3v38 · 2024-11-25
> >
> > Thank you for your replies but unfortunately they do not answer my comments satisfactorily. I keep my original vote.

---

### Official Review · Reviewer_M7oG · 2024-10-30

**Soundness:** 3
**Presentation:** 2
**Contribution:** 1
**Rating:** 1
**Confidence:** 2

**Summary:**

This work applies methods from topological data analysis to analyze representations in convolutional neural networks. They make some empirical observations such as differences in functional similarity across training epochs or data subsets using these TDA tools.

**Strengths:**

1. Many measurements of TDA metrics across several models, which led to some interpretable observations (e.g. when classes per training subset are very different, or when accuracies are different).

**Weaknesses:**

1. Use of k-means and other heuristic choices make analysis overly-complex and potentially dependent on arbitrary choices. See also 2.
2. Lack of "baselines". Instead of using heuristics to decrease computational cost, and instead of using these various methods that require some machinery to deal with, there are several simple and efficient baselines that could be compared against. For instance, simple statistics of activations (mean, variance, quantiles) or simple Euclidean or nearly-Euclidean (e.g. earth-mover) distances between activations would be natural to test against.
3. Analysis limited to a few old convolutional architectures on an image classification task.
4. Low on novelty. This work applies existing TDA methods to simple NNs, and does not find particularly strong empirical observations.  Others have also applied TDA methods to analyze NN representations.
5. Exposition could be organized better. Much of the main paper (e.g. training and test loss curves, experimental details, TDA background) is a bit out of place or does not flow well, and could benefit from moving some of the content to the appendix.

**Questions:**

What is the thesis? I am curious as to why you find topological data analysis tools to be promising for analyzing NNs.

---

> ### Author Response · Authors · 2024-11-19
> **Answer to Questions**
>
> Thank you for taking the time to review our submission.
>
> Our study hinges on the idea that analyzing the difference between the global structure of neural networks' functional graphs will lead us to better understand how networks represent information differently based on their architecture and optimization schemes. Knowing what these differences are, based on the controlled variables, can give us insight on how to build networks in order to achieve certain objectives. TDA then is our tool of choice to extract the underlying global information of the functional manifolds.

---

> > ### Comment · Reviewer_M7oG · 2024-11-20
> >
> > Thank you for your response. I will maintain my score of 1, as you have not addressed my concerns.

---

### Official Review · Reviewer_vMhX · 2024-11-01

**Soundness:** 2
**Presentation:** 1
**Contribution:** 1
**Rating:** 3
**Confidence:** 3

**Summary:**

This paper investigates the use of topological data analysis (TDA) to understand the global structure of deep neural networks (DNNs), particularly convolutional neural networks (CNNs). It focuses on tools such as persistent homology and Betti curves to analyze and compare DNNs across training epochs and disjoint subsets of the ImageNet dataset. The study explores how different CNN architectures, trained on subsets of data, exhibit distinct functional graph structures as they learn, and demonstrates that Betti curve similarity can effectively measure these structural differences. Results show that Betti curve similarity can differentiate DNN models, potentially providing insights into model behaviors and offering a new framework for theoretical analysis of DNN architectures

**Strengths:**

The strengths of this paper lie in its innovative application of TDA to examine deep neural networks, particularly CNNs. By leveraging tools like persistent homology and Betti curve similarity, it opens new pathways for understanding the global structural properties of DNNs beyond conventional metrics such as accuracy.

**Weaknesses:**

1.	The font size in figures is too small.
2.	There is too much preliminaries, and many of them are unnecessary or can be put in the appendix. Since this is an empirical paper, I am expecting more empirical results.
3.	The motivation is unclear. The authors should at least illustrate what motivates them to use TDA to measure the similarity of DNNs.
4.	The practical meanings of the proposed method should be tested, not just stated as future work.

**Questions:**

1.	Why the use of TDA to measure the similarity is necessary? For the same task, how does other similarity measures perform?
2.	Aren't there any related works?

---

> ### Author Response · Authors · 2024-11-19
> **Answers to Questions**
>
> Thank you for taking the time to review our submission.
>
> 1) We employee TDA to measure similarity since topological descriptors can give us a global summary of the functional graphs and their underlying manifolds. We currently do not test against other similarity measures but this will be an excellent addition to our analysis.
>
> 2) Yes there are related works, the most relevant to our study being Corneanu et. al. _What Does It Mean to Learn in Deep Networks? And, How Does One Detect Adversarial Attacks?_ This is cited as [3] in our paper. We can consider adding a sections dedicated to related works.

---

> > ### Comment · Reviewer_vMhX · 2024-11-26
> >
> > Thank you for your replies but unfortunately they do not answer my comments satisfactorily. I keep my original vote.

---

### Official Review · Reviewer_tcu6 · 2024-11-01

**Soundness:** 2
**Presentation:** 2
**Contribution:** 3
**Rating:** 5
**Confidence:** 3

**Summary:**

This paper applies tools from Topological Data Analysis (TDA), specifically persistent homology and Betti curve similarity, to analyze the global structure of deep neural networks' (DNNs). By examining the functional graphs of convolutional neural networks (CNNs) trained on disjoint subsets of ImageNet, the authors aim to provide a framework for comparing neural network architectures across datasets and epochs. The empirical experiments over across architectures and datasets demonstrate that PH or Betti curve can

**Strengths:**

1. **Novel Approach**: Using TDA tools to analyze neural networks across datasets and epochs is an intriguing approach, especially in trying to capture global behavior via Betti curves.
2. **Experiment setup**: Comprehensive experimental setup with multiple architectures and datasets.

**Weaknesses:**

* What specific insights do Betti curves provide that other network analysis methods (like SHAP) do not? The study mentions detecting changes in internal representations but does not clarify how these changes impact performance or model understanding.
   - [SHAP] https://arxiv.org/abs/1705.07874
* The study provides valuable insights into using TDA for neural network analysis, though it would benefit from stronger theoretical foundations and broader architectural exploration. The practical implications for network design and analysis could also be more thoroughly discussed.
* The K-means++ algorithm is just K-means algorithm with random initialization, and it is still performs linear partitioning of the space. The preliminary anlaysis of the clusters (not well-separated) might be due to the inherent non-linearity in the latent space.

**Questions:**

* The paper trains CNNs on 30 subsets of ImageNet but does not demonstrate whether the Betti curve similarity has practical utility beyond distinguishing models. What actionable insights can practitioners derive from the similarity scores?
* Given the heavy computational requirements, how does this method scale to larger models beyond CNNs? Could the approach work for other computationally heavy models?
* How sensitive are the topological features to different initialization scheme?
* The link to the implementation code at page 1 is not working.

---

> ### Author Response · Authors · 2024-11-19
> **Answers to Questions**
>
> Thank you for taking the time to review our submission.
>
> 1) Actionable insights are a topic of further research, but thus far we are confident that the similarity measure itself can be used to: i) create ensembles of networks that are topologically diverse, ii) provide part of a framework for model distillation where we conserve the topological structure of the network, and iii) create ablation studies where we can measure differences in activation topologies across architecture geometry.
>
> 2) Yes, however, we inevitably must reduce the number of points in a functional graph in order to compute the Betti curves. Some architectures lend themselves better to such reductions than others, e.g., ViT vs. CNNs.
>
> 3) Very good question. We don't have a clear answer at this time, but this is yet another avenue for future research. Essentially the initialization scheme, the optimizer, the learning rate, etc. could all have an impact on the topology of the functional graph.
>
> 4) All links that could be used to identify us have been redacted for the review process.

---

> ### Comment · Reviewer_tcu6 · 2024-11-26
>
> Dear authors,
>
> Thank you for your replies. The concerns raised in questions and weaknesses have not been fully addressed, and I unfortunately keep my original score.

---

### Official Review · Reviewer_W4qc · 2024-11-04

**Soundness:** 1
**Presentation:** 1
**Contribution:** 1
**Rating:** 1
**Confidence:** 4

**Summary:**

The paper proposes some method for comparing activations  of CNN by first reducing the set of all neurons to its k-means++ clusters and then applying to the resulting weighted graph with 1000 vertices the Betti curves comparison.  VGG-16, ResNET-18 and two small CNNs are used in experiments. It is demonstrated empirically that the  proposed method can distinguish the activations of the 4 models evaluated at different epochs.

**Strengths:**

The proposed  research direction of comparing functional circuits using topological tools is interesting

**Weaknesses:**

Here is a list of some of principal weaknesses of the paper:

1.Dependency on dimensionality reduction techniques. The k-means++ clustering technique is employed to reduce the dimensionality of neuron activations. However, this method may introduce approximation errors and fails to capture structure accurately, which could impact the persistent homology (PH) and Betti curve results. The reliance on this non-linear dimensionality reduction method limits the accuracy and interpretability of the results. It is not clear what is the significance to the concrete functioning of neural nets of the dissimilarity/similarity of 2- or 3-dimensional Betti numbers on two correlation graphs of such k-means clusters.

2.Ambiguity in quantifying functional differences. The Betti curve similarity metric provides a quantitative comparison of DNNs'  structures but does not offer fine-grained insight into how specific layers or components contribute to observed differences. This lack of layer-specific interpretability makes it challenging to apply these findings in model optimization or architecture refinement.

3.Limited generalizability across model architectures. The study focuses on a few convolutional neural network (CNN) architectures (LeNet, AlexNet, VGG-16, and ResNet-18). The findings may not generalize to other types of neural networks, such as transformers or recurrent networks, which differ significantly in structure and functional graph working.

4.The phrasing of principal mathematical concept is somewhat misleading. The persistence diagram is not "visual representation of Betti numbers of the complex as a function of the filtration parameter" as it contains a lot more information than just this,  since two complexes can have the same Betti numbers at each value of filtration parameter but have at the same time essentially different persistence diagrams.

5.Practical applications for neural networks architecture choice or neural networks training are absent.

6.An access to the source code is not provided for reproducibility check purposes. Despite mentioning in the paper, the provided link is broken and leads to an error page.

7.The writing can be improved. The introductory material on persistence homology is too lengthy and should probably be included in the appendix. While the results section presents data, including Betti curve similarities across models, subsets, and epochs, the discussion could be expanded to explore implications in greater depth. This includes connecting findings back to practical applications or limitations more thoroughly, such as how these topological insights might practically impact model selection, training.

**Questions:**

What are the practical applications of the proposed method?

---

> ### Comment · Reviewer_W4qc · 2024-12-01
> **Official comment**
>
> As the pointed issues were not addressed, I keep the initial score

---

### Meta-Review · Area_Chair_Xaz7 · 2024-12-19

**Metareview:**

**Summary**:
This paper advocates the use of some standard tools from topological data analysis (TDA) for understanding the global structure of deep neural networks by comparing activations, focusing particularly on CNNs.  Specifically, k-means++ is employed to cluster the neurons, and Betti curves are used to analyse the resulting functional subgraphs. Experiments were also presented for  VGG-16, ResNet-18 and two other CNN architectures.

**Strengths**:
The reviewers generally appreciated the novelty of the proposed approach, acknowledging the method introduced a useful metric that complements the conventional metrics such as accuracy. The clarity of presentation, the experimental setup, and the overall methodology, were also acknowledged.

**Weaknesses**:
The reviewers raised several concerns, including, unclear impact of dimensionality reduction, limited generalisability across architectures, lack of stronger theoretical foundations, missing related work, and absence of ablation analysis.

**Recommendation**:
The overall response to this work in the current form was rather lukewarm with more than one reviewer recommending strong reject. The authors did not respond at all to some reviewers, and provided limited responses to others, leaving most of the concerns unresolved. Therefore, I am unable to recommend this work for ICLR 2025. I hope the authors can take into account the reviewers' feedback to improve their work.

**Additional Comments On Reviewer Discussion:**

Please see the Metareview above for all the relevant details.

---

### Decision · Program_Chairs · 2025-01-22

Reject